# FuseNorm: Achieving the Best of Both Worlds from PreNorm and PostNorm

## Abstract

The success of Large Language Models (LLMs) hinges on the stable training of deep Transformer architectures. A critical design choice is the placement of normalization layers, leading to a fundamental trade-off: the "PreNorm" architecture ensures training stability at the cost of potential performance degradation in deep models, while the "PostNorm" architecture offers strong performance but suffers from severe training instability. In this work, we propose FuseNorm, a novel technique designed to resolve this dilemma by integrating the strengths of both paradigms. FuseNorm adopts the clean residual path of PreNorm to stabilize signal propagation while employing a PostNorm-style computation that normalizes the output of the residual connection, thereby enhancing model performance. We provide a theoretical analysis demonstrating that FuseNorm, combined with a principled scaling strategy, maintains bounded signal variance throughout the network, preventing the gradient issues that plague PostNorm models, and alleviating the representation collapse of PreNorm. Empirically, FuseNorm consistently outperforms standard normalization schemes in both dense and Mixture-of-Experts (MoE) scenarios, paving the way for more powerful and stable Transformer architectures.

## 1 Introduction

The Transformer architecture (Vaswani et al., 2017) has become the cornerstone of modern natural language processing, demonstrating unprecedented success and scalability, particularly in the realm of Large Language Models (LLMs) (Brown et al., 2020; Touvron et al., 2023). The remarkable capabilities of these models are not merely a function of their attention mechanisms but are also deeply rooted in fundamental design choices that ensure stable and effective training at scale. Among these, the interplay between residual connections (He et al., 2016) and normalization layers (Lei Ba et al., 2016) stands out as a critical factor for success. While both are integral, our work focuses on advancing the understanding and application of normalization within deep Transformer networks.

The evolution of normalization in Transformers provides a compelling narrative of balancing performance and stability. The original Transformer design employed a "PostNorm" architecture, where Layer Normalization (LN) is applied after the residual connection. This configuration is often associated with strong performance in shallower models but is quite difficult to train as network depth increases, suffering from vanishing and exploding gradients that prevent convergence beyond a few dozen layers (Wang et al., 2019; Xiong et al., 2020). To overcome this limitation, the "PreNorm" architecture was proposed, which moves the normalization layer to the input path of each sub-layer, before the residual addition. This simple yet effective change ensures a clean, unnormalized residual path, which significantly stabilizes gradient flow and enables the training of models with tens of layers (Wang et al., 2019).

Consequently, the PreNorm Transformer has become the de-facto standard for training deep LLMs (Touvron et al., 2023; Liu et al., 2024; Yang et al., 2025). However, this stability can come at a cost, as PreNorm models sometimes exhibit a "depth degradation" problem, where performance does not improve, or even degrades with increased depth (Wang et al., 2024; Li et al., 2020; Liu et al., 2020; Sun et al., 2025). This issue is frequently attributed to the tendency of PreNorm architectures to optimize an ensemble of shallower subnetworks, an effect facilitated by their direct residual paths. While effective to a point, this ensemble approach eventually hits a performance ceiling as network depth increases. Conversely, PostNorm models are believed to learn more potent, deeply integrated

features but are notoriously hampered by training instability. This phenomenon has spurred a new wave of research. On one hand, efforts have been made to stabilize PostNorm training to harness its performance benefits in deeper networks (Shleifer et al., 2021; Liu et al., 2020; Wang et al., 2024; Zhuo et al., 2025). On the other hand, researchers are exploring ways to mitigate the performance limitations of PreNorm architectures (OLMo et al., 2024; Sun et al., 2025).

In this work, we introduce FuseNorm, a novel normalization technique that directly addresses this trade-off. Instead of choosing between the two paradigms or attempting to fix one, FuseNorm is designed to marry the performance benefits of PostNorm with the training stability of PreNorm. Concretely, FuseNorm adopts the core design principle of PreNorm by establishing a clean path between layers, while simultaneously following the PostNorm computational workflow that normalizes the residual of the layer's input and output. A comparative analysis of gradient dynamics reveals the advantages of FuseNorm over both PreNorm and PostNorm.

To further ensure robust training for large-scale models, we also outline principles for an effective scaling strategy, encompassing both depth and width scaling. Our analysis demonstrates that FuseNorm, when paired with appropriate initialization and learning rate schedules, effectively preserves signal norms as information propagates through the network. We specifically show that the gradient signals avoid the vanishing gradient problem. Our experimental results validate this stability, demonstrating that FuseNorm provides a solid foundation for training Transformers with dozens of layers and billions of parameters. This holds true for both standard dense architectures and sparse Mixture-of-Experts (MoE) models, ensuring that performance consistently improves with scale. Concretely, in a 5B dense model and a 16B MoE model (with 2.4B active parameters), FuseNorm outperforms the standard Pre-LN approach by an average of 1.8 and 1.2 points, respectively, on the LM Harness evaluation benchmark. Our contributions demonstrate that it is possible to achieve the "best of both worlds," of PostNorm and PreNorm, paving the way for more powerful and efficient Transformer architectures.

## 2 RELATED WORK

The placement of Layer Normalization (LN) is critical to Transformer stability. The original PostNorm design applies normalization after the residual connection, behaving effectively in shallower models, but suffering from unstable gradients in deep networks. A simple yet impactful modification, PreNorm, reverses this order by applying normalization to the input of the sub-layer, maintaining a "clean" residual path that mitigates vanishing gradients (Wang et al., 2019) and removes the need for aggressive warm-up schedules (Xiong et al., 2020; Liu et al., 2020). This stability advantage made it the standard in Vision Transformer (ViT) (Dosovitskiy et al., 2021), GPT-3 (Brown et al., 2020), and most modern large language models (Touvron et al., 2023; Yang et al., 2025; Team et al., 2025).

However, deep PreNorm models exhibit their own limitations. The clean residual path can lead to representation collapse, where upper layers fail to learn new features (Li et al., 2020). This "Curse of Depth" is characterized by exponential growth in activation variance and layers that devolve into identity functions, limiting the benefits of scaling depth (Sun et al., 2025).

These challenges spurred two research streams. The first focuses on fixing PreNorm. Normformer (Shleifer et al., 2021) adds extra LNs to balance gradient norms, while LayerNorm Scaling (Sun et al., 2025) explicitly controls activation variance by scaling the LN output. The second stream aims to stabilize PostNorm to harness its performance. Zhang et al. (2019) showed that a depth-scaled initialization strategy could alleviate the gradient vanishing problem, and Huang et al. (2020) later provided theoretical support, demonstrating that with careful initialization, deep PostNorm models can be optimized effectively. Methods like DeepNorm (Wang et al., 2024) have successfully trained thousand-layer Transformers using specific rescaling and initialization strategies, highlighting PostNorm's sensitivity to these choices.

More recently, several hybrid approaches have emerged to combine the benefits of both paradigms. Mix-LN (Li et al., 2025) adopts an inter-layer hybrid strategy, using PostNorm for shallower layers and transitioning to PreNorm for deeper layers. In contrast, HybridNorm (Zhuo et al., 2025) utilizes an intra-layer approach, employing a QKV-Norm (where normalization is applied on the head dimension) before the attention computation while retaining a PostNorm after the attention output's residual connection, thereby attempting to gain stability without sacrificing performance.

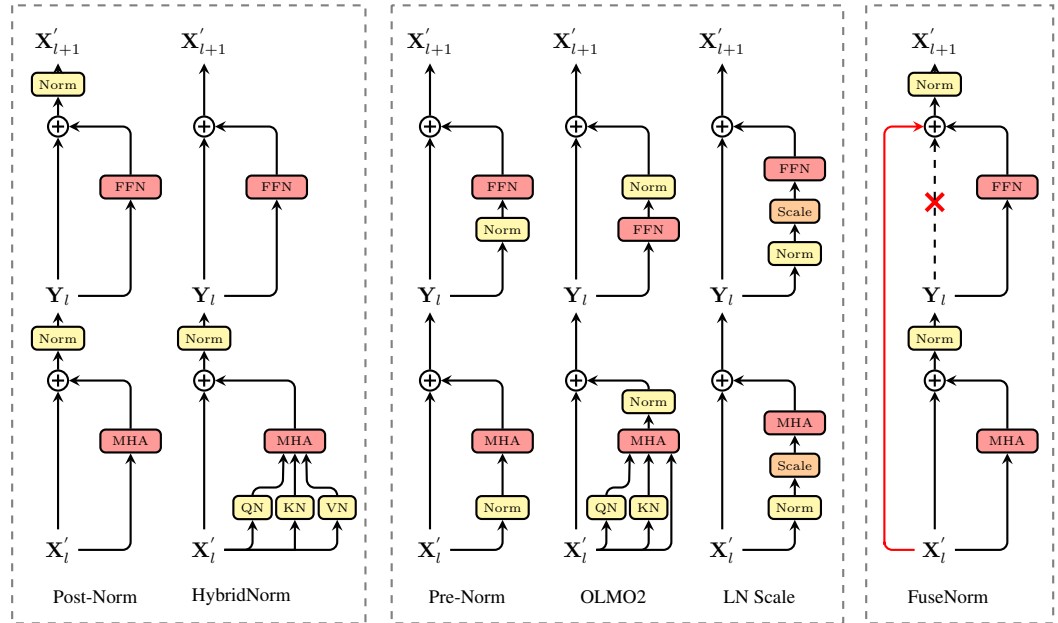

Figure 1: Comparisons of our proposed FuseNorm with PostNorm, PreNorm, and other advanced variants. Here, we take the dense model as an instance, and MHA denotes the multi-head attention, FFN denotes the feedforward network. Note that MHA can also be replaced by GQA, MLA and other attention variants. When switching to the MoE models, FFN could be replaced by MoE modules.

## 3 FUSENORM

In this section, we introduce FuseNorm, a novel modification to the Transformer block architecture designed to enhance training stability and performance. We begin by reviewing the standard PreNorm and PostNorm structures, and then present the FuseNorm architecture and provide a theoretical analysis of its advantages, focusing on its benefit against the PreNorm and PostNorm models. The overall architecture of FuseNorm and other variants is shown in Figure 1.

### 3.1 PRELIMINARIES: NORMALIZATION IN TRANSFORMERS

The placement of layer normalization within a Transformer block is a critical design choice that significantly impacts model training dynamics and final performance. The two dominant strategies are PostNorm and PreNorm.

**Post-LayerNorm (PostNorm):**  The original Transformer architecture introduced the PostNorm variant. In this setup, LayerNorm is applied after the residual connection in each sub-layer. The computation flow for a single block is:

$$\mathbf{Y}_l = \text{LN}(\text{MHA}(\mathbf{X}_l^{'}) + \mathbf{X}_l^{'}) \tag{1}$$

$$\mathbf{X}_{l+1}^{'} = \text{LN}(\text{FFN}(\mathbf{Y}_l) + \mathbf{Y}_l) \tag{2}$$

where $\mathbf{Y}_l$ denotes the residual output of the first sub-layer (Multi-Head Attention, short for MHA) and the layer input $\mathbf{X}_l^{'}$. $\mathbf{X}_{l+1}^{'}$ denotes the layer output. While PostNorm often yields strong model performance and regularization, it is difficult to train in very deep networks. The gradients can vanish or explode, necessitating careful learning rate warm-up and initialization strategies.

**Pre-LayerNorm (PreNorm):**  To address the training instability of PostNorm, especially when training deep models, the PreNorm variant was proposed. Here, LayerNorm is applied to the input of each sub-layer, before the residual connection. This creates a clean, identity-based residual path. The

computation flow is:

$$\mathbf{Y}_l = \text{MHA}(\text{LN}(\mathbf{X}_l^{'})) + \mathbf{X}_l^{'} \tag{3}$$

$$\mathbf{X}_{l+1}^{'} = \text{FFN}(\text{LN}(\mathbf{Y}_l)) + \mathbf{Y}_l \tag{4}$$

## 3.2 THE FUSENORM ARCHITECTURE

The goal of FuseNorm is to combine the performance benefits of PostNorm with a structural design that promotes the training stability characteristic of PreNorm. We achieve this by modifying the information flow in the second residual connection of a PostNorm block.

Given the input to the block $X'$, the computation proceeds as follows:

$$\mathbf{Y}_l = \text{LN}(\text{MHA}(\mathbf{X}_l^{'}) + \mathbf{X}_l^{'}) \tag{5}$$

$$\mathbf{X}_{l+1}^{'} = \text{LN}(\text{FFN}(\mathbf{Y}_l) + \mathbf{X}_l^{'}) \tag{6}$$

The first sub-layer (e.g., MHA) follows the standard PostNorm structure. However, in the second sub-layer (Feed-Forward Network, short for FFN), the residual connection adds the original block input, $\mathbf{X}_l^{'}$, instead of the intermediate representation, $\mathbf{Y}_l$. This minor change creates a powerful direct skip connection from the input to the output of the entire block.

We make a specific modification for the first layer of the network ($l = 1$). The input to this layer consists of raw word embeddings, $\mathbf{E}$, which are not normalized, unlike the inputs to subsequent layers. To ensure the MHA sub-layer operates on a normalized input and maintains a consistent attention pattern across the network, we apply an additional LayerNorm to the embeddings before the MHA computation. However, this initial LayerNorm is not applied to $\mathbf{E}$ within the identity path of the residual connection, thereby preserving a clean, unmodified "highway" for information to flow seamlessly. Thus, the forward pass for the first layer is defined as:

$$\mathbf{Y}_1 = \text{LN}\left(\text{MHA}(\text{LN}(\mathbf{E})) + \mathbf{E}\right) \tag{7}$$

$$\mathbf{X}_2^{'} = \text{LN}\left(\text{FFN}(\mathbf{Y}_1) + \mathbf{E}\right) \tag{8}$$

## 3.3 COMPARATIVE ANALYSIS OF GRADIENT DYNAMICS

The design of FuseNorm strategically combines the structural strengths of PreNorm and PostNorm. In this section, we provide a theoretical analysis of gradient dynamics, demonstrating how FuseNorm mitigates the specific pathologies of its predecessors: (1) the representation collapse characteristic of deep PreNorm networks, and (2) the vanishing gradients inherent to PostNorm architectures.

**Advantage over PreNorm: Preventing Deep-Layer Representation Collapse** A primary pathology in deep PreNorm Transformers is the uncontrolled growth of feature variance, which leads to representation collapse. In a standard PreNorm block, the main residual path acts as an identity map, causing variance to accumulate linearly with depth. Formally, for a network of depth $L$, the variance of the hidden states scales as $\text{Var}(X_L') = \Theta(L)$ (Kedia et al., 2024).

This variance explosion degrades the learning capability of deep layers. Consider the Jacobian of the $l$-th PreNorm block. Let $\text{Res}(\cdot)$ denote the transformation within the residual branch (e.g., Self-Attention or FFN). Since the transformation within the residual branch operates on inputs normalized by the feature standard deviation $\sigma_l = \sqrt{\text{Var}(X_l')}$, the Jacobian $J_{\text{Pre}}$ can be expressed as:

$$J_{\text{Pre}} = \mathbb{I} + \underbrace{\frac{\partial \text{Res}(X_l')}{\partial X_l'} \cdot \mathcal{O}\left(\frac{1}{\sigma_l}\right)}_{\text{Vanishing Term}} \tag{9}$$

As $l \to \infty$, $\sigma_l \to \infty$, causing the residual term to vanish at a rate of $\mathcal{O}(1/\sqrt{l})$. Consequently, $J_{\text{Pre}} \to \mathbb{I}$, meaning the gradients effectively bypass the transformative sub-layers. The deep layers thus devolve into identity functions, contributing minimal new information to the representation.

*The FuseNorm Solution.* FuseNorm circumvents this issue by enforcing a variance reset at the end of each block (Eq. 6). By applying LayerNorm to the combined output of the FFN and the skip

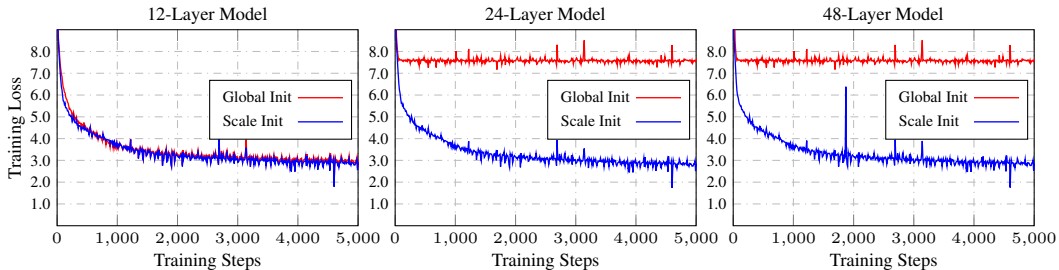

Figure 2: **Early-stage training stability analysis. Setup:** We train dense models with a fixed hidden dimension $d = 1536$ across increasing depths (12, 24, 48 layers) for 5000 steps to evaluate the impact of initialization on stability.

connection, FuseNorm ensures that the output variance remains bounded, i.e., $\text{Var}(X'_{l+1}) = \Theta(1)$. This prevents the Jacobian collapse, forcing deep layers to contribute meaningfully to the feature transformation.

**Advantage over PostNorm: Enhancing Stability by Mitigating Gradient Decay.** While standard PostNorm models maintain bounded variance, they suffer from exponential gradient decay due to the serial application of normalization layers on the main propagation path. To formalize the advantage of FuseNorm, we analyze the signal attenuation factor under a simplified homogeneous setting.

**Assumption 1** (Homogeneous Variance). *We assume that the pre-normalized sums in each sub-layer exhibit a consistent standard deviation $\sigma > 1$ across the network.*

Under Assumption 1, the gradient magnitude $\|G\|$ passing through a normalization layer is scaled by a factor of $1/\sigma$. We compare the cumulative scaling behavior at depth $L$:

*PostNorm Decay.* In a PostNorm block, the signal passes through two normalization layers sequentially (one after Attention, one after FFN). The gradient norm at layer $L$ is proportional to the product of these scaling factors:

$$\|G_{\text{Post}}(L)\| \propto \left(\frac{1}{\sigma} \cdot \frac{1}{\sigma}\right)^L = \sigma^{-2L} \tag{10}$$

*FuseNorm Preservation.* In contrast, FuseNorm's topology (Eq. 6) introduces a single normalization on the aggregate block output $X'_{l+1}$. The gradient signal is effectively scaled only once per block regarding the macro-architecture: $\|G_{\text{Fuse}}(L)\| \propto \left(\frac{1}{\sigma}\right)^L = \sigma^{-L}$.

Comparing the decay rates reveals a fundamental advantage:

$$\|G_{\text{Post}}(L)\| \propto \|G_{\text{Fuse}}(2L)\| \tag{11}$$

This derivation implies that regarding gradient magnitude preservation, a FuseNorm network of depth $L$ is dynamically equivalent to a PostNorm network of depth $L/2$. For deep networks (e.g., $L = 50$) with $\sigma \approx 1.1$, this theoretical difference results in gradient signals for FuseNorm that are orders of magnitude larger than those of PostNorm ($1.1^{50} \approx 117\times$), effectively eliminating the vanishing gradient problem that prevents PostNorm convergence.

## 4 STABLE SCALING

Despite its PreNorm-inspired skip connection for gradient stability, FuseNorm's PostNorm-style output reintroduces a degree of sensitivity to hyperparameters. Our empirical results confirm that to prevent early-stage instability and fully leverage its performance advantages, FuseNorm requires initialization and learning rate parameterization more aligned with those for PostNorm models. This necessitates a principled approach to scaling.

To formalize this, we first abstract the core training process. The output of a linear layer, $\mathbf{Y}$, depends on the final weights, which are the sum of the initial weights $\mathbf{W}_{init}$ and the accumulated updates

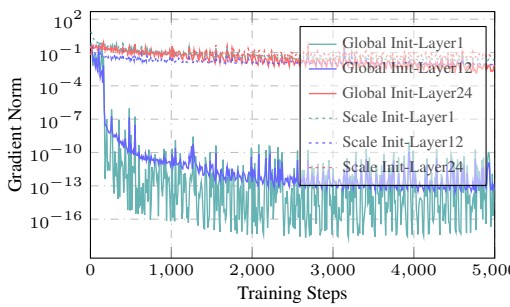 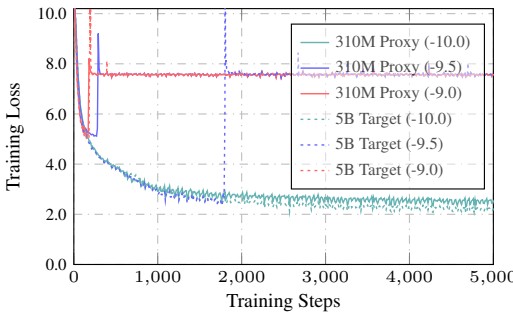

Figure 3: **Gradient norm dynamics (Analysis of the 24-layer model in Figure 2).** We visualize layer-wise gradient norms for the **exact 24-layer configuration** that exhibited instability. (See Figure 13 for the full visualization of all 24 layers).

Figure 4: **Validation of width scaling stability transfer.** We compare a 310M proxy (hidden dimension $d = 640$, solid lines) and a 5B target ($d = 2560$, dotted lines) under a fixed depth ($L = 64$).

from the optimizer, $\Delta\mathbf{W}$:

$$\mathbf{Y} = \mathbf{X}\left(\mathbf{W}_{init} + \Delta\mathbf{W}(\mathcal{G}, \eta)\right) \tag{12}$$

where the update term $\Delta\mathbf{W}$ is a function of the gradients $\mathcal{G}$ and the learning rate $\eta$. This conceptual model highlights the two critical levers we must control for stable and predictable scaling: the starting point of the optimization, defined by the initial weights ($\mathbf{W}_{init}$), and the dynamics of the updates, governed by the learning rate ($\eta$).

## 4.1 DEPTH SCALING: INITIALIZATION FOR GRADIENT STABILITY

To ensure stable training as network depth increases, we analyze the conditions for stable initialization. The stability is governed by the spectral norm of the layer-wise Jacobian matrix, which must remain close to 1 to prevent exploding or vanishing gradients. Our analysis of the proposed FuseNorm architecture is summarized in the following theorem.

**Theorem 1** (Condition for Stable Depth Scaling[1]). *For a FuseNorm Transformer model with $L$ layers to maintain stable training dynamics as $L \to \infty$, the output variance of its residual sub-layers (e.g., the FFN block $\mathcal{F}$) must scale inversely with the total depth. Specifically, the condition is :*

$$Var(\mathcal{F}(\mathrm{LN}(\mathbf{x}))) = \mathcal{O}(1/L) \tag{13}$$

**Practical Initialization Strategy.** Theorem 1 offers a clear initialization strategy. To directly meet the variance condition $\mathrm{Var}(\mathcal{F}(\cdot)) = \mathcal{O}(1/L)$, initializing the FFN block parameters is crucial. We recommend setting the standard deviation of the output matrix $\mathbf{W}_2$'s initialization to $\mathcal{O}(1/\sqrt{L})$, which effectively reduces the matrix norm and ensures the FFN's output variance aligns with $\mathcal{O}(1/L)$. Similarly, the attention output matrix $\mathbf{W}_O$ should be scaled to preserve gradient integrity within the FFN block and support the near-identity principle of deep residual networks. Thus, we apply $\mathcal{O}(1/\sqrt{L})$ scaling to both $\mathbf{W}_2$ and $\mathbf{W}_O$. Notably, this theoretical approach matches the initialization method used in frameworks like Megatron-LM. While we apply this initialization globally to ensure fairness, our experiments in Section 5 demonstrate that PostNorm still fails to converge at scale, validating our analysis in Section 3.3 that the superior stability of FuseNorm stems from its architectural topology rather than specific initialization strategies.

**Experimental Validation.** To verify the effectiveness of this scaled initialization, we conducted experiments on models with 12, 24, and 48 layers, as shown in Figure 2. We observed that a standard global initialization leads to training collapse when scaled to 24 layers, while the scaled initialization allows stable training even at 48 layers. Analyzing the gradient dynamics of the failing 24-layer model, as depicted in Figure 3, reveals that the collapse is caused by severe shallow-layer gradient vanishing, which our strategy successfully prevents.

---

[1]The detailed proof, which involves bounding the spectral norm of the Jacobian matrix, is provided in Appendix A.

Table 1: Our FuseNorm results against PreNorm (Touvron et al., 2023) on various configurations. All models are trained on the same subset of the SlimPajama dataset (from 30B to 200B) with the Mistral tokenizer Jiang et al. (2023). The last column shows the average over all benchmarks that use (normalized) accuracy as the metric.

| Model | Param | Tokens | Wiki. ppl↓ | LMB. ppl↓ | LMB. acc↑ | PIQA acc↑ | Hella. acc_norm↑ | SciQ acc↑ | ARC-c acc_norm↑ | Wino. acc↑ | Avg. acc↑ |
|---|---|---|---|---|---|---|---|---|---|---|---|
| | | | | | *Dense Models* | | | | | | |
| PreNorm | 740M | 30B | 22.5 | 22.9 | 39.5 | 66.3 | 39.3 | 79.1 | 25.3 | 49.8 | 49.8 |
| FuseNorm | 740M | 30B | 20.6 | 26.3 | 38.5 | 67.9 | 42.4 | 80.1 | 25.1 | 50.8 | 50.8 |
| PreNorm | 1.3B | 100B | 17.4 | 12.6 | 51.1 | 70.0 | 49.6 | 84.4 | 26.8 | 53.1 | 55.8 |
| FuseNorm | 1.3B | 100B | 15.7 | 12.0 | 52.1 | 72.3 | 54.0 | 85.5 | 28.8 | 55.3 | 58.0 |
| PreNorm | 5B | 200B | 12.5 | 6.9 | 61.4 | 73.7 | 64.4 | 90.7 | 34.4 | 60.1 | 64.1 |
| FuseNorm (w/ WS) | 5B | 200B | **11.7** | 6.4 | 63.9 | **75.8** | **67.2** | 91.6 | 35.2 | 61.5 | 65.9 |
| FuseNorm (w/o WS) | 5B | 200B | 11.8 | **5.8** | **64.2** | 75.7 | 66.9 | **92.0** | **36.3** | **64.2** | **66.6** |
| | | | | | *MoE Models* | | | | | | |
| PreNorm | 16B-A2.4B | 200B | 12.0 | 6.7 | 61.7 | 75.9 | 66.0 | 89.1 | 37.0 | 60.9 | 65.1 |
| FuseNorm | 16B-A2.4B | 200B | **11.4** | **6.5** | **63.0** | **76.2** | **68.1** | **90.8** | **38.6** | **61.3** | **66.3** |

Table 2: Comparisons with other advanced LN variants.

| Model | Param | Tokens | Wiki. ppl↓ | LMB. ppl↓ | LMB. acc↑ | PIQA acc↑ | Hella. acc_norm↑ | SciQ acc↑ | ARC-c acc_norm↑ | Wino. acc↑ | Avg. |
|---|---|---|---|---|---|---|---|---|---|---|---|
| PostNorm | 5B | 200B | | | | | failed | | | | |
| PreNorm | 5B | 200B | 12.5 | 6.9 | 61.4 | 73.7 | 64.4 | 90.7 | 34.4 | 60.1 | 64.1 |
| HybridNorm (lr=$1e^{-4}$) | 5B | 200B | 12.2 | 6.5 | 62.7 | 75.1 | 66.2 | 91.4 | 35.4 | 61.3 | 65.4 |
| Mix-LN | 5B | 200B | 12.3 | 6.9 | 61.3 | 75.6 | 65.1 | 90.1 | 34.1 | 61.4 | 64.6 |
| LayerNorm Scale | 5B | 200B | 13.6 | 14.9 | 51.0 | 73.1 | 60.7 | 86.1 | 32.2 | 58.3 | 60.2 |
| OLMO2 | 5B | 200B | 14.0 | 9.9 | 53.7 | 73.1 | 59.2 | 87.4 | 30.6 | 57.2 | 60.2 |
| Peri-LN (lr=$1e^{-4}$) | 5B | 200B | 12.3 | 7.2 | 59.5 | 75.7 | 65.1 | 90.3 | 35.2 | 59.9 | 64.3 |
| FuseNorm | 5B | 200B | **11.8** | **5.8** | **64.2** | 75.7 | **66.9** | **92.0** | **36.3** | **64.2** | **66.6** |

## 4.2 WIDTH SCALING: PREDICTIVE STABILITY TRANSFER

While we prioritize depth scaling for structural stability, handling width scaling is equally critical for large-scale training. We empirically confirm that FuseNorm follows the width scaling protocols proposed by Everett et al. (2024). Crucially, this adherence facilitates the predictive stability transfer observed in our experiments. As illustrated in Figure 4, stability thresholds identified on a 310M proxy model accurately predict the behavior of a 5B target model. Results confirm a robust predictive relationship: learning rates that fail on the proxy also fail on the target, while those stable on the proxy remain stable on the target. This stable transfer enables the use of efficient proxies to reliably identify safe hyperparameters, eliminating the need for risky tuning on large models. This is of paramount importance in industrial-scale model training, where avoiding catastrophic training divergence due to the prohibitive economic and temporal costs associated with failure.

## 5 EXPERIMENTS

**Experimental Setups** We summarized the detailed setups of selected baselines, model configurations, training hyperparameters and the corresponding dataset/evaluation in Appendix B.

**Results on LM Evaluation Harness** We conducted a series of experiments across various model sizes, utilizing both dense and MoE architectures. The comprehensive results, presented in Table 1, demonstrate a clear and consistent performance advantage for our approach when compared to the PreNorm baseline. Note that we report FuseNorm results under two configurations: "w/ WS" (with Width Scaling) employs our proposed scaling strategy to adjust the learning rate, while "w/o WS" (without Width Scaling) strictly adopts the hyperparameters identical to the PreNorm baseline for a controlled comparison. Specific training setups and hyperparameter details are provided in Appendix B. Specifically, our method yields an average improvement of 1.0-2.5 points across different

Table 3: Deep scaling evaluation: Comparisons with advanced LN variants on a 128-layer model.

| Model | Param | Tokens | Wiki. ppl ↓ | LMB. ppl ↓ | LMB. acc ↑ | PIQA acc ↑ | Hella. acc_norm ↑ | SciQ acc ↑ | ARC-c acc_norm ↑ | Wino. acc ↑ | Avg. |
|---|---|---|---|---|---|---|---|---|---|---|---|
| PostNorm | 6.5B | 400B | | | | | failed | | | | |
| PreNorm | 6.5B | 400B | 11.9 | 6.6 | 62.6 | 75.3 | 65.2 | 90.5 | 32.6 | 60.1 | 64.4 |
| HybridNorm | 6.5B | 400B | | | | | failed | | | | |
| Mix-LN | 6.5B | 400B | 11.4 | 6.2 | 64.0 | 76.1 | 67.5 | 90.4 | 35.7 | 62.5 | 66.0 |
| FuseNorm | 6.5B | 400B | **10.4** | **5.5** | **65.4** | **77.5** | **70.6** | **92.6** | **37.7** | **66.6** | **68.4** |

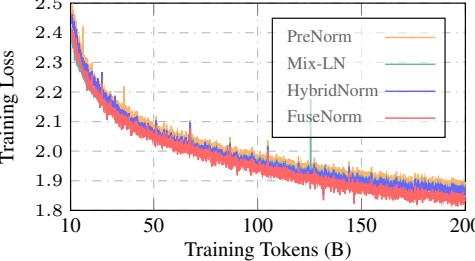

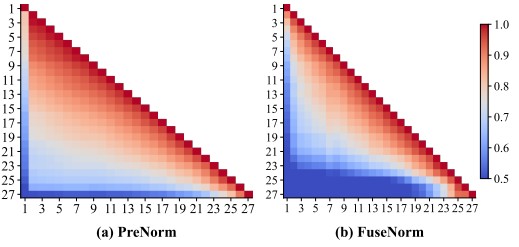

(a) PreNorm    (b) FuseNorm

Figure 5: Training loss comparison with advanced normalization variants (Dense-5B model).

Figure 6: Cosine similarity between the outputs of each pair of layers for MoE-2.4B-16B model.

dense model capacities (from 740M to 5B). Similarly, our FuseNorm can also achieve an average improvement of 1.2 when switching to the MoE setting, further demonstrating its effectiveness. Notably, our design also addresses a common optimization challenge by enabling stable training of PostNorm models (failed to train), effectively eliminating the gradient vanishing.

**Comparisons with Other Variants** To comprehensively understand the advantages of FuseNorm, we also compared it with several recent variants, as shown in Table 2. To ensure a fair comparison, we re-implemented these methods, including Mix-LN (Li et al., 2025), HybridNorm (Zhuo et al., 2025), LayerNorm Scale (Sun et al., 2025), OLMO2 (OLMo et al., 2024) and Peri-LN (Kim et al., 2025) in our codebase using Megatron and conducted experiments on a 5B dense model trained on 200B tokens for robust results. We summarize the LM Harness evaluation results in Table 2 and plot the training curves in Figure 5. Unexpectedly, OLMO2 and LayerNorm Scale yielded worse results than the PreNorm baseline with the same hyperparameters. For LayerNorm Scale, we attribute this phenomenon to its scaling factor, which is strongly dependent on the total model depth. In models with large depth (e.g., 64), the gradients of shallower layers may be scaled down too much, which negatively impacts the optimization process. Meanwhile, Peri-LN, HybridNorm and Mix-LN (PostNorm ratio $\alpha = 25\%$) deliver better performance than PreNorm, and our FuseNorm consistently outperforms these advanced variants in both training loss and downstream performance.

**Scalability on Ultra-Deep Networks.** We stress-test stability by scaling to 128 layers (6.5B parameters). Table 3 and Figure 9 focuses exclusively on the most competitive variants from Table 2, excluding those that already underperformed the PreNorm. Results reveal that success at moderate depths does not guarantee scalability: notably, HybridNorm failed to converge in this deep regime despite its earlier promise. In contrast, FuseNorm not only maintained robust stability but also achieved a substantial gain over PreNorm. This validates FuseNorm's unique capacity for "stable transfer", preventing catastrophic training collapses in industrial-scale scaling.

## 6 EMPIRICAL ANALYSIS

In this section, we provide empirical validation for the theoretical advantages of our FuseNorm architecture discussed in section 3.3.

**Empirical Validation of Preventing Representation Collapse.** To empirically validate the issue of representation collapse in PreNorm and assess whether FuseNorm can mitigate this problem, we

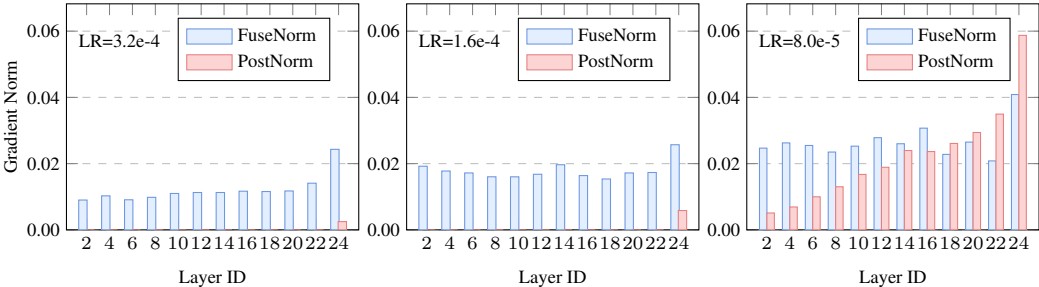

Figure 7: Comparison of FFN weight gradient norms between our FuseNorm model and standard PostNorm using the 740M model configuration (detailed in Appendix B) across three learning rate conditions. **(Left and Middle)** At a high learning rate ($> 8 \times 10^{-5}$), PostNorm training collapses, showing near-zero gradients in all but the final layer, while FuseNorm remains stable. **(Right)** At a lower learning rate ($8 \times 10^{-5}$), PostNorm suffers significant shallow-layer gradient decay, whereas FuseNorm maintains balanced gradients across all layers, demonstrating superior training stability and healthier gradient propagation.

examine the evolution of its internal representations. The gradient analysis suggests that the Jacobian of a deep PreNorm block converges to an identity matrix, which implies that deeper layers fail to learn new features, causing the output of one layer to approximate the next. Figure 6 presents the cosine similarity between the hidden state outputs across all layer pairs in the MoE-A2.4B-16B model, reveals that the contrast between the architectures is most pronounced in deeper layers: while PreNorm retains high similarity, FuseNorm evolves towards markedly distinct representations. This observation is quantitatively confirmed by Figure 12, where PreNorm maintains a stagnant, flat curve ($> 0.5$) across depths, whereas FuseNorm exhibits a rapid, near-linear decay down to $\approx 0.25$. These results demonstrate that FuseNorm effectively drives long-range feature evolution, ensuring that deep layers contribute meaningful information unlike the redundant layers in PreNorm.

**Empirical Validation of Mitigated Gradient Decay.** We empirically validate the severe gradient decay issue in PostNorm and examine whether FuseNorm can alleviate this problem. Our experiments are conducted on a 24-layer model with 740 million parameters. A comparison with a standard PostNorm model across three learning rates demonstrates the distinct advantages of FuseNorm, as illustrated in Figure 7. At a low learning rate ($8 \times 10^{-5}$), FuseNorm sustains well-balanced gradients in contrast to the skewed distribution observed in PostNorm, which indicates pronounced shallow-layer decay. As the learning rate increases, PostNorm undergoes catastrophic collapse, whereas FuseNorm maintains stability. Notably, FuseNorm consistently exhibits a balanced gradient distribution with minimal variance between layers across all learning rate settings. This empirical evidence suggests that the healthy gradient propagation achieved by FuseNorm contributes to more robust training dynamics and extends the stable learning rate range.

To further corroborate this finding from a spectral perspective, we provide an analysis of the eigenvalue distribution and spectral utilization of the feature representations in Appendix C. These analyses confirm that FuseNorm maintains a healthier singular value spectrum compared to baselines, effectively utilizing the full dimensionality of the feature space.

## 7 CONCLUSIONS

In this work, we addressed the long-standing trade-off between training stability and model performance in Transformer architectures. We introduced FuseNorm, a novel normalization strategy that synergistically combines the stability of the PreNorm design with the performance benefits of the PostNorm computation. Our theoretical analysis confirmed that FuseNorm ensures stable signal propagation, enabling the training of exceptionally deep and large models without encountering the gradient issues that typically hinder PostNorm architectures. Empirically, FuseNorm consistently outperforms existing normalization methods across various model scales. By resolving a critical bottleneck in LLM training, our work paves the way for the development of more powerful, efficient, and scalable models.

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

# A PROOF OF THEOREM 1

*Proof.* The forward pass for the $l$-th layer of our FuseNorm architecture is given by:

$$\mathbf{Y}_l = \text{LN}\left(\text{MHA}(\mathbf{X}'_l) + \mathbf{X}'_l\right) \tag{14}$$

$$\mathbf{X}'_{l+1} = \text{LN}\left(\text{FFN}(\mathbf{Y}_l) + \mathbf{X}'_l\right) \tag{15}$$

The stability of backpropagation depends on the spectral norm of the Jacobian matrix, $\left\|\frac{\partial \mathbf{X}'_{l+1}}{\partial \mathbf{X}'_l}\right\|_2$. Let $\mathbf{Z}_l = \text{FFN}(\mathbf{Y}_l) + \mathbf{X}'_l$, $\mathbf{A}_l = \text{MHA}(\mathbf{X}'_l) + \mathbf{X}'_l$. By the chain rule and the submultiplicativity of spectral norms, we have:

$$\left\|\frac{\partial \mathbf{X}'_{l+1}}{\partial \mathbf{X}'_l}\right\|_2 \leq \left\|\frac{\partial \text{LN}(\mathbf{Z}_l)}{\partial \mathbf{Z}_l}\right\|_2 \left\|\frac{\partial \mathbf{Z}_l}{\partial \mathbf{X}'_l}\right\|_2 \tag{16}$$

The spectral norm of the Layer Normalization Jacobian is approximately the inverse of the standard deviation of its input, $\|\frac{\partial \text{LN}(\mathbf{Z}_l)}{\partial \mathbf{Z}_l}\|_2 \approx \frac{1}{\sigma_{\mathbf{Z}_l}}$. The second term can be bounded using the triangle inequality (where FFN is denoted as $\mathcal{F}$):

$$\left\|\frac{\partial \mathbf{Z}_l}{\partial \mathbf{X}'_l}\right\|_2 = \left\|\mathbb{I} + \frac{\partial \mathcal{F}(\mathbf{Y}_l)}{\partial \mathbf{X}'_l}\right\|_2 \leq 1 + \left\|\frac{\partial \mathcal{F}(\mathbf{Y}_l)}{\partial \mathbf{X}'_l}\right\|_2 \tag{17}$$

The term $\left\|\frac{\partial \mathcal{F}(\mathbf{Y}_l)}{\partial \mathbf{X}'_l}\right\|_2$ can be further decomposed using the chain rule and submultiplicativity:

$$\left\|\frac{\partial \mathcal{F}(\mathbf{Y}_l)}{\partial \mathbf{X}'_l}\right\|_2 = \left\|\frac{\partial \mathcal{F}}{\partial \mathbf{Y}_l}\frac{\partial \mathbf{Y}_l}{\partial \mathbf{X}'_l}\right\|_2 \leq \left\|\frac{\partial \mathcal{F}}{\partial \mathbf{Y}_l}\right\|_2 \left\|\frac{\partial \mathbf{Y}_l}{\partial \mathbf{X}'_l}\right\|_2 \tag{18}$$

In the context of feedforward neural networks (FFNs), the Jacobian norm, represented as $\|\frac{\partial \mathcal{F}}{\partial \mathbf{Y}_l}\|_2$, can be approximated by the expression $\|\frac{\partial \mathcal{F}}{\partial \mathbf{Y}_l}\|_2 \approx \|\mathbf{W}_1\mathbf{W}_2\|_2$. Here, $\mathbf{W}_1$ and $\mathbf{W}_2$ denote the weight matrices associated with the FFN's two linear layers, under the assumption of an identity activation function (Takase et al., 2023).

For the Jacobian of the first sub-layer, $\frac{\partial \mathbf{Y}_l}{\partial \mathbf{X}'_l}$, the formulation is more intricate. It can be bounded as $\|\frac{\partial \mathbf{Y}_l}{\partial \mathbf{X}'_l}\|_2 \approx \frac{1}{\sigma_{\mathbf{A}_l}}\|\mathbb{I} + \frac{\partial \text{MHA}}{\partial \mathbf{X}'_l}\|_2$. The Jacobian of the attention block, $\frac{\partial \text{MHA}}{\partial \mathbf{X}'_l}$, is further decomposed into the output projection matrix $\mathbf{W}_O$ and the preceding multi-head attention mechanism, which encompasses queries, keys, values, and the softmax operation. We denote the Jacobian of this attention mechanism as $\mathbf{J}^{\mathcal{A}'}$. Consequently, we have $\|\frac{\partial \text{MHA}(\mathbf{X}'_l)}{\partial \mathbf{X}'_l}\|_2 = \|\mathbf{W}_O\mathbf{J}^{\mathcal{A}'}\|_2$.

Combining these results, we obtain the final upper bound on the Jacobian's spectral norm:

$$\left\|\frac{\partial \mathbf{X}'_{l+1}}{\partial \mathbf{X}'_l}\right\|_2 \lesssim \frac{1}{\sigma_{\mathbf{Z}_l}}\left(1 + \frac{\|\mathbf{W}_2\mathbf{W}_1\|_2(1 + \|\mathbf{W}_O\mathbf{J}^{\mathcal{A}'}\|_2)}{\sigma_{\mathbf{A}_l}}\right) \tag{19}$$

For stable training through $L$ layers, the cumulative product of these Jacobians must not vanish or explode. This is critically dependent on the outer scaling factor $\frac{1}{\sigma_{\mathbf{Z}_l}}$ at each layer. For the product to remain of order $\mathcal{O}(1)$ as $L \to \infty$, it is required that $\sigma_{\mathbf{Z}_l} = 1 + \mathcal{O}(1/L)$.

We analyze the variance of $\mathbf{Z}_l$. By the definition of LayerNorm, the input $\mathbf{X}'_l$ has unit variance, i.e., $\text{Var}(\mathbf{X}'_l) = 1$. The input to the FFN, $\mathbf{Y}_l$, is similarly normalized. Under the standard initialization assumption that the residual branch $\mathcal{F}(\mathbf{Y}_l)$ and the identity path $\mathbf{X}'_l$ are uncorrelated, the variance of their sum is the sum of their variances:

$$\text{Var}(\mathbf{Z}_l) = \text{Var}(\mathbf{X}'_l) + \text{Var}(\mathcal{F}(\mathbf{Y}_l)) = 1 + \text{Var}(\mathcal{F}(\mathbf{Y}_l)) \tag{20}$$

For the condition $\sigma_{\mathbf{Z}_l} = 1 + \mathcal{O}(1/L)$ to hold, the variance must be $\text{Var}(\mathbf{Z}_l) = (1 + \mathcal{O}(1/L))^2 = 1 + \mathcal{O}(1/L)$. Comparing this with our derived variance, we arrive at the central condition:

$$\text{Var}(\mathcal{F}(\mathbf{Y}_l)) = \mathcal{O}(1/L) \tag{21}$$

This completes the proof.

$\square$

## B    EXPERIMENTAL SETUPS

**Baseline**    We evaluate our proposed FuseNorm on both dense and MoE settings. The PreNorm and PostNorm are two major baselines, and others like LayerNorm Scaling (Sun et al., 2025), HybridNorm (Zhuo et al., 2025), Mix-LN (Li et al., 2025), and OLMO2 (OLMo et al., 2024) are stronger baselines that we have comprehensively compared.

**Model Configurations**    We mainly evaluate our dense models across three configurations as follows: (1) 740M: The hidden dimension is 1536, and the intermediate size of FFN is 4224. The head is 24, with each head dimension being 64. The 740M is a 24-layer model. (2) 1.3B: The hidden dimension is 1536, and the intermediate size of FFN is 4224. The head is 24. We scaled the depth to twice that of the 740M model, which leads to a 48-layer model. (3) 5B: Our largest size model has a hidden dimension of 2560, an intermediate size of 6912, where 40 heads and a depth of 64. For the MoE model, we use the Deepseek-V3 small-scale model (Liu et al., 2024), activated by 2.4 billion parameters within approximately 16 billion parameters in total. It employs an MLA architecture, activating 6 out of 64 experts. The model consists of a total of 27 layers, with a hidden dimension of 2048.

**Training Hyperparameters**    We employ the AdamW optimizer (Kingma & Ba, 2015) with hyperparameters set to $\beta_1 = 0.9$ and $\beta_2 = 0.95$, and weight decay as 0.1. We set the max sequence length to 2K during pretraining with 200B tokens uniformly sampled from SlimPajama. As for the learning rate schedule, we first linearly increase the learning rate from 0 to the peak value, e.g., $2e^{-4}$, during the first 200 steps. Then the learning rate decays following a cosine curve until meeting the minimum learning rate predefined. Our batch size is set to 512K for a 740M model, 1M for a 1.3B model, and 4M for a 5B model. The gradient clipping norm is set to 1.0. We chose the Megatron initialization (Shoeybi et al., 2019) (referred to as 'Scale Init' in Section 4.1) as our default strategy for all models, including the baselines (PostNorm, PreNorm, HybridNorm, etc.). This isolates the impact of the architecture from the initialization scheme..

To ensure a fair and rigorous comparison in the 5B dense model experiments in Table 2, we carefully calibrated the configurations for each model:

- Standard Learning Rate ($2 \times 10^{-4}$): We adopted this commonly used learning rate for the PreNorm baseline. Additionally, OLMO2, LayerNorm Scale, and Mix-LN were also trained using this standard $2 \times 10^{-4}$ learning rate. Note that for Mix-LN, while the learning rate was kept standard, we performed a trade-off analysis on the architecture, finding that a configuration of 16 PostNorm layers and 48 PreNorm layers yielded the lowest training loss.

- Tuned Learning Rate ($1 \times 10^{-4}$): Certain baselines required lower learning rates for stability or optimal performance. For HybridNorm, we observed severe gradient vanishing with learning rates of $2 \times 10^{-4}$ and $1.5 \times 10^{-4}$; thus, we report the optimal stable result at $1 \times 10^{-4}$. Similarly, for Peri-LN, we empirically found that a $1 \times 10^{-4}$ learning rate yielded significantly better convergence and performance than $2 \times 10^{-4}$ for the Gemma-2 style architecture.

- FuseNorm Configurations: In Table 1, we present two settings for FuseNorm. "w/o WS" denotes the setting where FuseNorm uses the exact same hyperparameters as the PreNorm baseline (i.e., learning rate = $2 \times 10^{-4}$) for a direct comparison. "w/ WS" indicates the application of our Width Scaling strategy, which prescribed a learning rate of $2.4 \times 10^{-4}$ (selected based on the discrete search intervals of our proxy model).

For the deep scaling experiments involving the 128-layer 6.5B dense model, to keep a fair comparison, we search for the best learning rate on PreNorm models, and set $1.2 \times 10^{-4}$ as the default setting for all other models (including Mix-LN, HybridNorm, PostNorm and FuseNorm).

Regarding the MoE experiments in Table 1, we utilized identical configurations for both the PreNorm baseline and our FuseNorm model, setting the learning rate to $2 \times 10^{-4}$.

**Dataset and Evaluation**    Following the setup in previous studies, we randomly sampled 200B tokens from SlimPajama dataset which is widely available in the open-source literature. To evaluate

the performance of LLMs, we report the results from the LM Harness evaluation, including Lambada, PIQA, HellaSwag, SCIQ, ARC-c, and WinoGrande.

## C   SPECTRAL ANALYSIS OF FEATURE REPRESENTATIONS

**Analysis on Eigenvalue Distribution**    Following the setup in Loshchilov et al. (2025)'s work, we mainly plot the sorted eigenvalues of the input embedding as shown in Figure 8. We can observe that FuseNorm exhibits a much smaller range than PreNorm and HybridNorm. Interestingly, Mix-LN shows a similar trend to FuseNorm. Moreover, we also plot the condition number of MLP1 (up-projection matrix) and MLP2 (down-projection matrix) at each layer. The left part in Figure 10 shows the condition numbers for MLP matrices at different layer depths, and the right part summarizes the corresponding average condition number. Our FuseNorm exhibits the smallest condition numbers in the MLP1 matrix. When switching to the MLP2, we see the condition number is the smallest across all 4 models at the bottom 6 and top 15 layers. The observation here demonstrates the ability of FuseNorm to preserve a well-conditioned and expressive latent space throughout the entire network depth.

**Analysis on Spectral Utilization**    To quantify how effectively the FFN blocks utilize their high-dimensional latent space, we employ three diagnostic metrics: Hard Spectral Rank, Soft Spectral Rank, and the composite Effective Dimension Ratio (EDR) (Jha & Reagen, 2025), as visualized in Figure 11. We can observe that PreNorm suffers from severe spectral collapse, exhibiting the lowest utilization across all metrics (e.g., an average normalized HardRank of only 0.09). In contrast, FuseNorm demonstrates superior representational capability. For the dominant modes, FuseNorm achieves the highest average Hard Rank of 0.23, significantly outperforming the baselines. Regarding the spectral tail, FuseNorm ties for the top performance in Soft Rank (0.53), indicating it effectively prevents spectral dilution. Most notably, in terms of the Effective Dimension Ratio—which serves as a holistic measure of latent space efficiency—FuseNorm dominates with a score of 30.66, surpassing the second-best HybridNorm (27.12) and more than doubling the effective capacity of PreNorm (12.99). The layer-wise dynamics further reveal that FuseNorm maintains high and stable spectral utilization throughout the entire network depth. This empirical evidence demonstrates that FuseNorm successfully mitigates representational collapse, maximizing the expressivity of the learned features.

## D   ANALYSIS OF EXTREME DEPTH SCALING

In this section, we provide a detailed analysis of the "Stress-Testing" experiment presented in Figure 14, which evaluates the scalability and stability of FuseNorm under extreme depth conditions.

**Experimental Setup.**    To rigorously isolate the impact of network depth on training dynamics, we constructed a series of "tall and narrow" dense proxy models. We fixed the hidden dimension to a relatively small scale ($d = 384$) to reduce computational overhead while exponentially increasing the number of layers $L$ from 32 to 512 (specifically $L \in \{32, 64, 128, 256, 512\}$). All models were trained using our proposed Scale Init strategy with a fixed learning rate of $2 \times 10^{-4}$. This fixed learning rate setting serves as a stress test to verify whether the architecture can tolerate a unified optimization hyperparameter across vastly different depths without divergence.

**Overcoming Depth Degradation.**    A pervasive issue in deep Transformer training, particularly with PreNorm architectures, is "depth degradation" or the "curse of depth," where increasing the number of layers beyond a certain point yields diminishing returns or even performance regression due to representation collapse.However, as illustrated in Figure 14, FuseNorm exhibits a strict "deeper is better" trend. We observe a monotonic decrease in training loss as the depth doubles. Notably, the 512-layer model achieves the lowest training loss, demonstrating that FuseNorm effectively facilitates signal propagation through hundreds of layers, allowing the model to learn increasingly complex hierarchical features without saturating.

**Stability at Extreme Depths.**    Training a 512-layer Transformer is notoriously difficult for standard PostNorm architectures due to the exponential growth of gradients. Conversely, PreNorm often

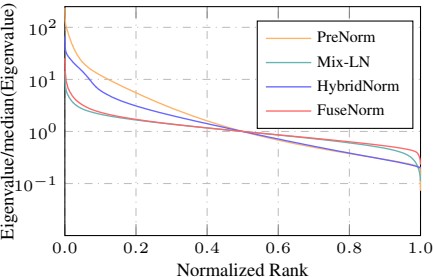

Figure 8: **Eigenspectrum Analysis of Input Embeddings.** We analyze the distribution of sorted eigenvalues normalized by their median value (log scale). While Mix-LN (teal) mitigates the extreme outliers seen in PreNorm, **we observe that it suffers from a rapid decay in the tail spectrum**, indicating potential dimensional collapse. In contrast, **we demonstrate that FuseNorm (red) maintains a significantly flatter and more uniform trajectory** across the entire rank, ensuring a more **isotropic representation space** with higher effective capacity.

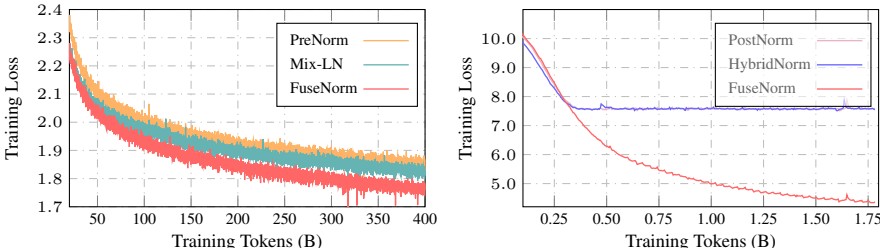

Figure 9: **Scalability analysis on a deep 128-layer dense model (6.5B parameters). Setup:** We evaluate training dynamics on a 128-layer, 6.5B parameter dense model trained for **400B tokens**. **Observation:** (Left) **FuseNorm (Red) significantly outperforms** stable baselines, achieving consistently lower training loss than PreNorm (Orange) and Mix-LN (Green) throughout the 400B token run. (Right) Deep scaling proves fatal for others: **PostNorm and HybridNorm fail to converge**, whereas FuseNorm remains robust. This confirms FuseNorm's capability to scale to exceptional depths and handle long-duration training.

suffers from identity mapping degeneracy at such depths. The successful convergence of the 512-layer FuseNorm model highlights the effectiveness of our theoretical contributions: (1) Gradient Stability: The combination of FuseNorm and Scale Init ensures that the gradient norms remain well-behaved even at $L = 512$, preventing the exploding gradients that typically plague PostNorm-style computations. (2) Representation Diversity: The continued reduction in loss suggests that the deep layers are actively contributing to the optimization process, rather than collapsing into identity functions as seen in deep PreNorm models. In summary, this stress test confirms that FuseNorm, backed by principled initialization, is robust enough to support large-scale scaling in the depth dimension, offering a viable path for training ultra-deep foundation models.

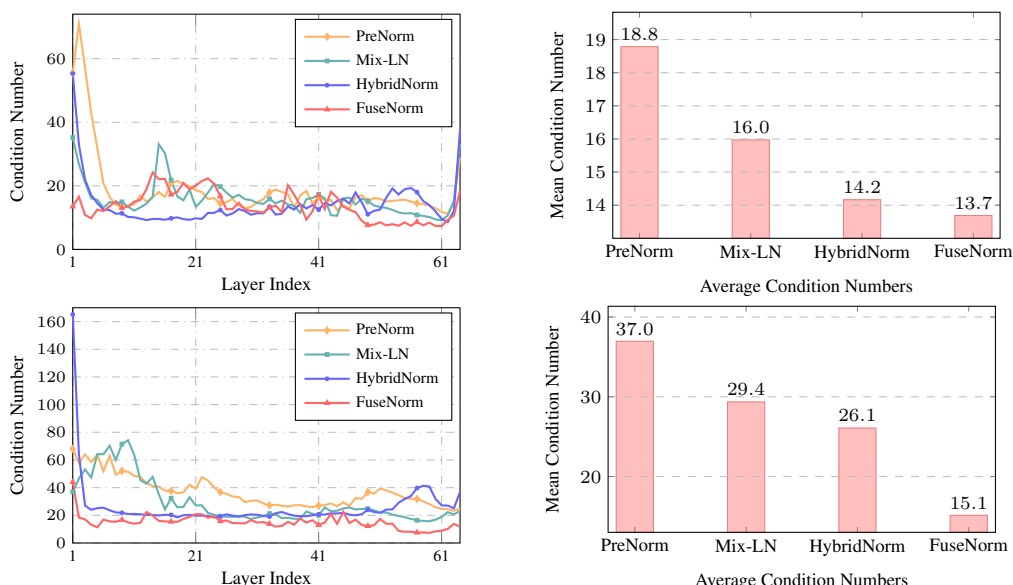

Figure 10: **Layer-wise condition number analysis of FFN weights.** We analyze the spectral properties of the **MLP1** (up-projection) and **MLP2** (down-projection) matrices across the network depth. **FuseNorm** demonstrates superior stability, maintaining significantly lower condition numbers, particularly in the **deeper layers**, which effectively prevents spectral degradation. In contrast, **PreNorm** exhibits elevated condition numbers in deeper blocks, reflecting the issue of *deep layer degeneration* where the representation capacity collapses. This comparison highlights FuseNorm's ability to preserve a well-conditioned and expressive latent space throughout the entire network depth.

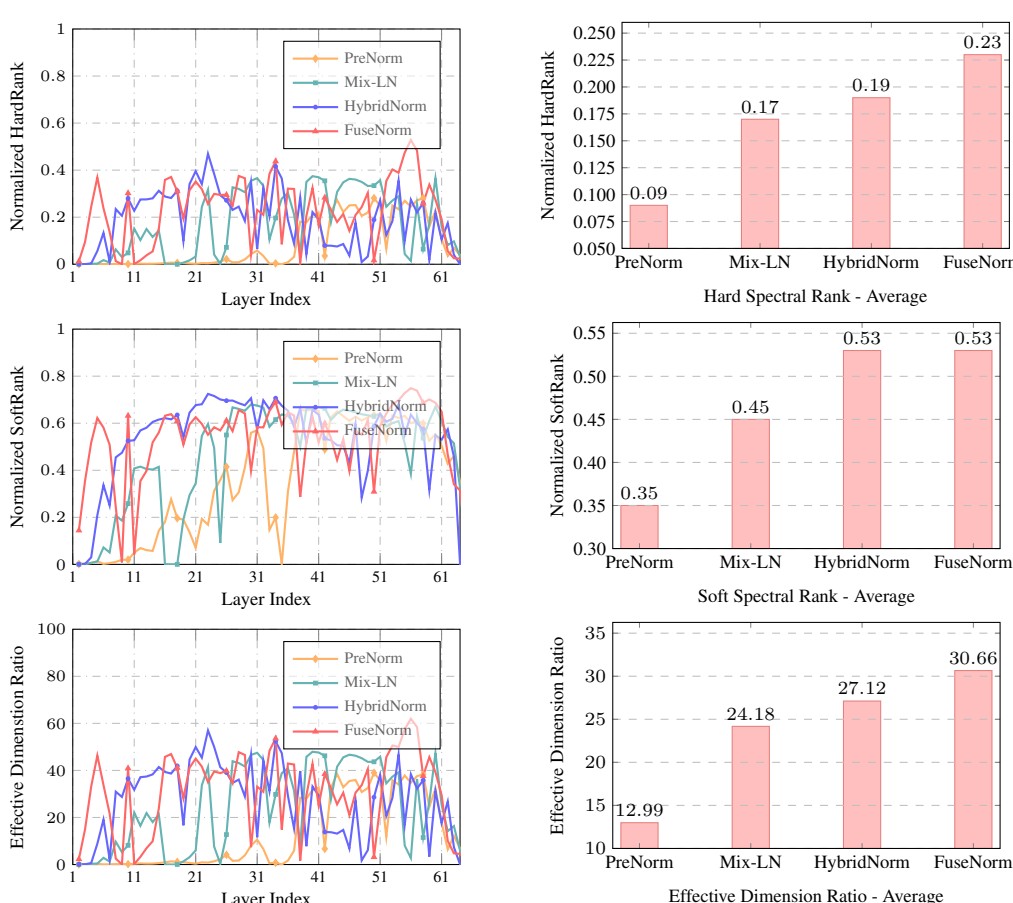

Figure 11: **FuseNorm achieves superior spectral utilization compared to other normalization baselines.** We evaluate the layer-wise dynamics (left) and average performance (right) of PreNorm, Mix-LN, HybridNorm, and FuseNorm across three spectral metrics. **(Top) Hard Spectral Rank:** FuseNorm demonstrates the strongest ability to preserve dominant eigen-directions, achieving the highest average rank of **0.23**, significantly outperforming HybridNorm (0.19) and the PreNorm baseline (0.09). **(Middle) Soft Spectral Rank:** FuseNorm effectively prevents spectral dilution, tying for the top performance with a score of **0.53**, ensuring a uniform distribution of information in the tail. **(Bottom) Effective Dimension Ratio:** Most notably, FuseNorm dominates in overall latent space efficiency with an effective ratio of **30.66**, surpassing the second-best HybridNorm (27.12) and more than doubling the capacity of PreNorm (12.99). Overall, FuseNorm proves to be the most effective scheme for mitigating spectral collapse and maximizing representational expressivity.

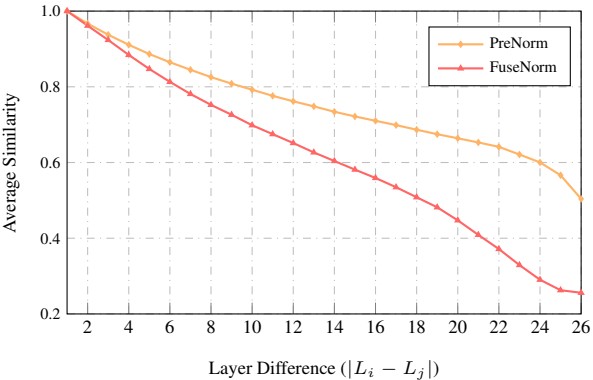

Figure 12: **Quantitative analysis of representation collapse (Complementing Figure 6).** To quantify the visual patterns observed in **Figure 6**, we aggregate the cosine similarity by layer distance on the MoE-2.4B-16B model. **Metric Calculation:** We compute the mean cosine similarity for all layer pairs $(L_i, L_j)$ grouped by their distance $k = |i - j|$. **Observation: PreNorm** exhibits a flat curve, maintaining high average similarity ($> 0.5$) even at the maximum layer distance ($k = 26$). This quantitatively confirms **representation collapse**, as deep layers fail to diverge from shallow ones. In contrast, **FuseNorm** shows a rapid, near-linear decay down to $\approx 0.25$. This significant drop demonstrates that FuseNorm drives **effective long-range feature evolution**, ensuring deep layers learn distinct representations.

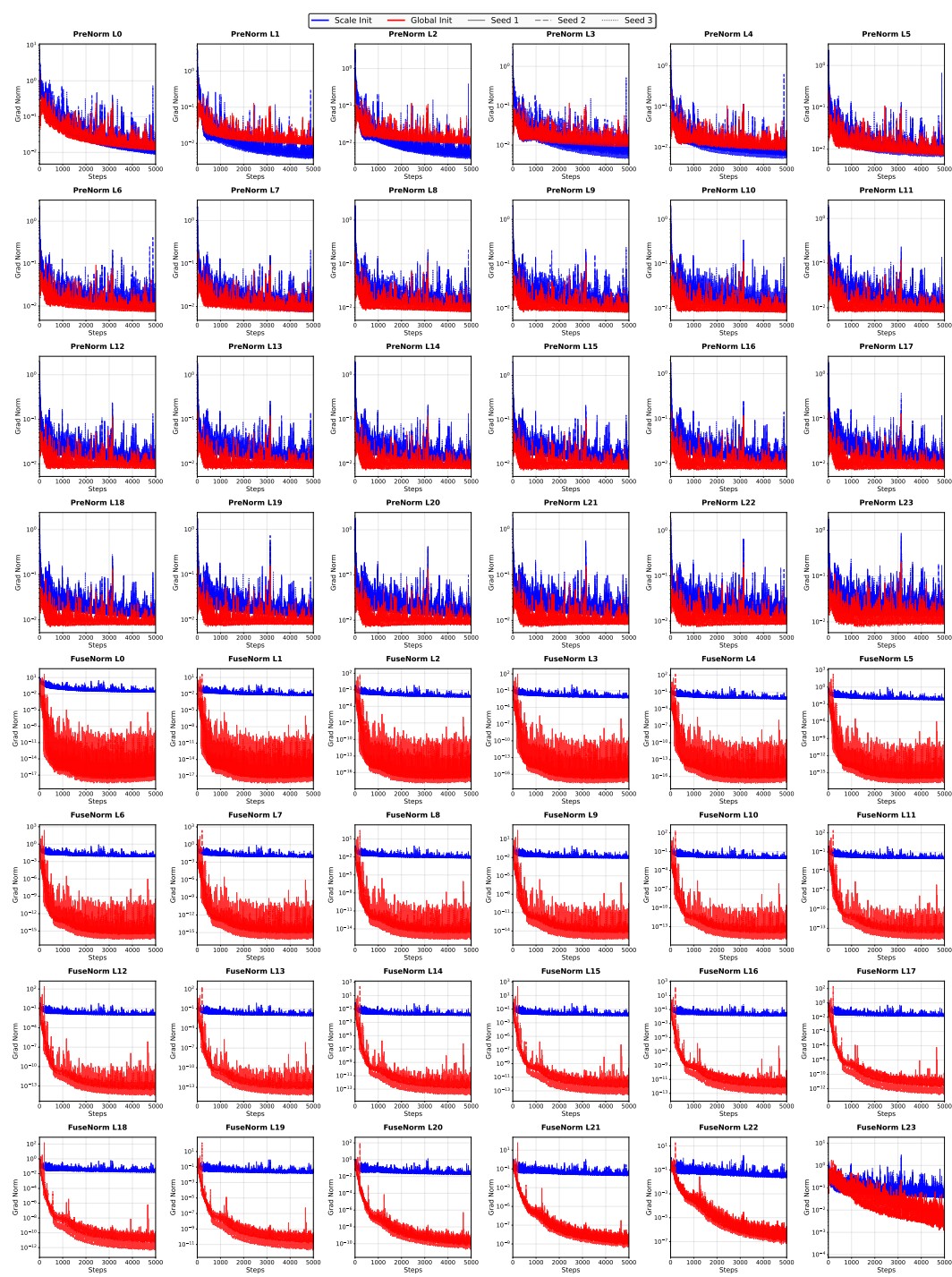

Figure 13: Comprehensive layer-wise gradient norm dynamics comparison (PreNorm vs. FuseNorm), which extends the original analysis of Figure 3 by visualizing gradient norms across all 24 layers for the first 5,000 training steps. We compare PreNorm (top rows) and FuseNorm (bottom rows) under two initialization strategies: the standard Global Init (Red) and our proposed Scale Init (Blue). The plots show the trend across 3 random seeds, with faint lines representing individual runs to demonstrate reproducibility. (1) Collapse without Scaling: The bottom rows reveal that FuseNorm with Global Init suffers from catastrophic gradient decay, where norms drop below $10^{-12}$, confirming the theoretical prediction of training collapse. (2) Effectiveness of Scale Init: Scale Init effectively stabilizes FuseNorm, maintaining healthy gradient norms ($\approx 10^{-1}$) comparable to the PreNorm baseline. (3) Baseline Comparison: PreNorm remains robust to initialization changes, but FuseNorm achieves similar stability only when paired with Scale Init.

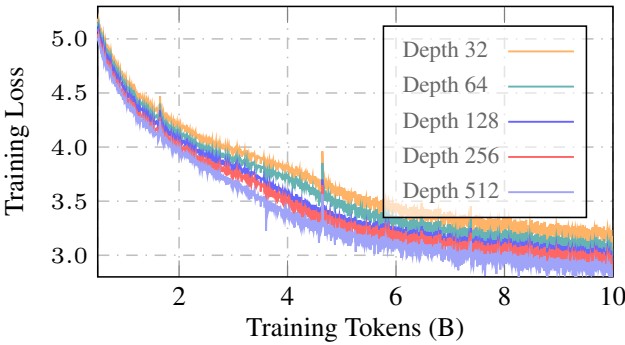

Figure 14: **Stress-testing "Deeper is Better".** Here we conducted a stress test on a base model (hidden dimension $d = 384$) with depths exponentially increasing from 32 to 512, all using a **fixed learning rate of** $2 \times 10^{-4}$ and Scale Init. The results show a clear **"deeper is better" trend**: training loss decreases monotonically as depth increases. This confirms that FuseNorm is not only stable at extreme depths (up to 512 layers) but also effectively **avoids the depth degradation problem** commonly observed in deep PreNorm models.

