# OpenReview forum: "FuseNorm: Achieving the Best of Both Worlds from PreNorm and PostNorm"
_ICLR.cc/2026/Conference — Submitted to ICLR 2026_

### Official Review · Reviewer_Deba · 2025-10-23

**Soundness:** 2
**Presentation:** 2
**Contribution:** 2
**Rating:** 2
**Confidence:** 4

**Summary:**

This paper addresses the trade-off between PreNorm and PostNorm Transformer architectures. PreNorm stabilizes training but often underperforms at scale. In contrast, PostNorm achieves better final performance but is prone to instability in deep networks. To reconcile these opposing properties, the authors propose FuseNorm, a simple yet effective modification that preserves the clean gradient path of PreNorm while applying Layer Normalization at the output like PostNorm.

**Strengths:**

The core ideas are well-motivated and supported by theoretical insights.

The method shows improvements in performance across various tasks and model scales.

**Weaknesses:**

The performance improvement is marginal, and in some cases (e.g., LMB perplexity, LMB accuracy, and ARC-c accuracy at 740M scale) the method underperforms compared to PreNorm. Moreover, on the 5B model, HybridNorm achieves better performance on ARC-c accuracy.

In addition, the paper lacks completeness in several aspects:

**Lack of comparison with relevant baselines**: The paper does not include a comparison with Peri-LN (Peri-LN: Revisiting Normalization Layer in the Transformer Architecture, ICML 2025), which is a closely related normalization method.

**Confounding factors in experimental design**: The method couples FuseNorm with Scale Init. Since initialization itself can significantly impact both training stability and final performance, it is unclear how much of the gain comes from FuseNorm vs. Scale Init. For example, a comparison with PreNorm + Scale Init would provide a clearer attribution of the source of improvement.

**Missing ablation experiments**: The paper does not provide standalone ablation experiments to disentangle the contribution of different architectural components (residual shortcut structure vs. normalization sequence) to training stability and performance.

**Reproducibility**: No implementation or code is provided, which makes it difficult to fully verify the empirical claims.

**Questions:**

Minor Errors

Line 505: In in -> In

Line 658: to twice as 740M -> to twice that of the 740M model

Line 659: maximize -> largest

Line 666: slimpajama -> SlimPajama

Line 674: Slimpajama -> SlimPajama

Line 676: Winograde -> WinoGrande

Line 676: Hellaswag -> HellaSwag

Line 675: LM harness evaluation -> the LM Harness evaluation

Line 174: LN(MHA(X′_l)) + X′_l) -> LN(MHA(X′_l) + X′_l)

Lines 175 and 190: The prime notation appears to be inconsistently rendered in mathematical expressions.

---

> ### Author Response · Authors · 2025-11-21
> **Response to the marginal improvement**
>
> Thank you for this observation. While the gain in any single configuration may vary, the consistent trend across our experiments shows a clear advantage for FuseNorm. We validated our method on both dense (from 740M to 5B) and MoE models. Although the improvement in the specific case you highlight is modest, FuseNorm achieves significant gains at larger scales, such as a **2.2 point average gain** on the 1.3B model and a **2.5 point gain**  on the 5B model. We would like to emphasize that these improvements are obtained with no increase in FLOPs compared to PreNorm, representing a valuable gain in efficiency.
>
> We performed an additional experiment on a challenging 6.5B, 128-layer configuration to investigate performance in extremely deep models. This experiment yielded several key findings:
>
> - FuseNorm scales successfully: Our method remained stable and showed notable performance gains, demonstrating its robustness at depth.
>
> - PreNorm degrades significantly: As hypothesized, PreNorm's performance collapsed in this 128-layer setup. This strongly supports our claim that it struggles with representation collapse in deeper architectures.
>
> - Other baselines struggle: HybridNorm failed to train at this depth. While Mix-LN did beat PreNorm, we attribute this to its shallower post-norm layers, which can be seen as a compromise between PreNorm and FuseNorm's designs.
>
> These results confirm that FuseNorm effectively mitigates the deep-model instability that plagues PreNorm, and its advantages become even more apparent as depth increases.
>
> | Model                      | Wiki.(ppl) | LMB.(ppl) | LMB.     | PIQA     | Hella.   | SCIQ     | ARC-c    | Winograde| Avg.     |
> | -------------------------- | ---------- | --------- | -------- | -------- | -------- | -------- | -------- | -------- | -------- |
> | PreNorm (lr=1.2e-4)        | 11.9       | 6.6       | 62.6     | 75.3     | 65.2     | 90.5     | 32.6     | 60.1     | 64.4     |
> | FuseNorm (lr=1.2e-4)       | 10.4       | 5.5       | 65.4     | 77.5     | 70.6     | 92.6     | 37.7     | 66.6     | 68.4     |
> | Mix-LN  (lr=1.2e-4)        | 11.4       | 6.2       | 64.0     | 76.1     | 67.5     | 90.4     | 35.7     | 62.5     | 66.0     |
> | HybridNorm (lr=1.2e-4)     | failed     | failed    | failed   | failed   | failed   | failed   | failed   | failed   | failed   |

---

> ### Author Response · Authors · 2025-11-21
> **Response to the comparisons with other baselines**
>
> We have already added the missing baselines all reviewer have raised. We hope the comparison now and the newly added more challenge setting could address your concerns! The results are summarized in the following table:
>
> | Model                      | Wiki.(ppl) | LMB.(ppl) | LMB.     | PIQA     | Hella.   | SCIQ     | ARC-c    | Winogrande| Avg.     |
> | -------------------------- | ---------- | --------- | -------- | -------- | -------- | -------- | -------- | --------  | -------- |
> | PreNorm (lr=2e-4)          | 12.5       | 6.9       | 61.4     | 73.7     | 64.4     | 90.7     | 34.4     | 60.1      | 64.1     |
> | FuseNorm (old)             | 11.7       | 6.4       | 63.9     | 75.8     | 67.2     | 91.6     | 35.2     | 61.5      | 65.9     |
> | FuseNorm (lr=2e-4)         | 11.8       | 5.8       | 64.2     | 75.7     | 66.9     | 92.0     | 36.3     | 64.2      | **66.6** |
> | Mix-LN (lr=2e-4)           | 12.3       | 6.9       | 61.3     | 75.6     | 65.1     | 90.1     | 34.1     | 61.4      | 64.6     |
> | HybridNorm (lr=1e-4)       | 12.1       | 6.5       | 62.7     | 75.1     | 66.2     | 91.4     | 35.4     | 61.3      | 65.4     |
> | LayerNorm Scale (lr=2e-4)  | 13.6       | 14.9      | 51.0     | 73.1     | 60.7     | 86.1     | 32.2     | 58.3      | 60.2     |
> | Peri-LN (lr=2e-4)          | 12.9       | 7.9       | 58.2     | 73.8     | 62.7     | 90.8     | 32.1     | 60.3      | 63.0     |
> | Peri-LN (lr=1e-4)          | 12.3       | 7.2       | 59.5     | 75.7     | 65.1     | 90.3     | 35.2     | 59.9      | 64.3     |
> | OLMO2 (lr=2e-4)            | 14.0       | 9.9       | 53.7     | 73.1     | 59.2     | 87.4     | 30.6     | 57.2      | 60.2     |
>
> For a fairer comparison, we follow the suggestion of Reviewer pvm3, using a learning rate of 2e-4 to rerun our FuseNorm. Here, we removed the width scaling strategy, which means all initializations are totally the same! Besides, we add the results of Peri-LN (which is Gemma2-style architectures, namely sandwich-like norm), OLMO2, and LayerNorm Scale to show a comprehensive comparison. Through the updated results, we summarized that:
>
> - Superior Performance of FuseNorm: Under these strictly controlled settings, FuseNorm (lr=2e-4) achieves the best overall performance. It reaches an average accuracy of 66.6, surpassing our originally reported results by 0.7 (stronger results on Lambada, SCIQ, ARC-C, and Winogrande). Notably, it significantly outperforms the PreNorm baseline by 2.5 average score, which is indeed significant.
> - Optimized Baselines for Fair Comparison: Our goal is to compare FuseNorm against the best possible version of each baseline. Therefore, we tuned the hyperparameters for the competing methods:
>     - HybridNorm: We observed gradient vanishing with learning rates of 2e-4 and 1.5e-4; thus, we report the optimal stable result at 1e-4.
>     - Mix-LN: We performed a trade-off analysis, finding that a configuration of 16 PostNorm layers and 48 PreNorm layers yielded the lowest training loss.
>     - Peri-LN: We found that a 1e-4 learning rate yielded significantly better convergence and performance than 2e-4 for the Gemma-2 style architecture.
> - Investigation of OLMO2 and LayerNorm Scale: We are currently conducting further hyperparameter tuning for OLMO2 and LayerNorm Scale. However, preliminary analysis suggests reasons for their current lower performance compared to PreNorm:
>     - LayerNorm Scale: As the model depth increases (64 layers), the learned scaling factors may become extremely small, potentially hindering signal propagation. The original paper did not validate this method at this depth, suggesting it may require specific tuning for deep networks.
>     - OLMO2: While we utilized the OLMO2 architecture (including QK-Norm and reorder-norm), we maintained the standard Megatron initialization for fairness, rather than the fixed standard deviation (0.02) used in their technical report. We are currently validating the impact of this initialization difference.
>
> - Overall: These updated experiments demonstrate that FuseNorm is robust and achieves the best performance when compared fairly against a wide range of optimized baselines.

---

> ### Author Response · Authors · 2025-11-21
> **Response to Confounding factors in experimental design and the missing ablation studies**
>
> > Confounding factors in experimental design:
>
> We want to draw your attention that our reported result of PreNorm is just PreNorm + Scale Init (Depth Scaling). All baselines here used the depth scaling strategy to ensure a fair comparison. In our previous manuscript, the use of width scaling is mainly to make sure that FuseNorm can be successfully optimized when scaling to a larger setting, not from the perspective of improving the performance. Our new results in the above table have shown that FuseNorm with depth scaling only can achieve better performance and beat the PreNorm baseline by a larger gap.
>
> > Missing ablation experiments:
>
> Thank you for raising this issue! We would like to clarify that, from the architectural perspective, our design is a simple but effective choice to marry the stability of PostNorm with the optimization benefits of PreNorm in the most direct way possible. So far, we have treated the residual connection as a fixed constraint required for gradient flow. And through our preliminary experiments, we did not find any other good variants that can achieve comparable performance with FuseNorm. We think the proposed FuseNorm is not just the result of simple enumerations!
>
> As the results in the above table, we can see that FuseNorm trained without the width scaling (in this way, FuseNorm used the same initialization with PreNorm and other variants), can achieve even better performance (66.6 vs. 65.9).
>
> We are also looking forward to your further advice to better distinguish each contribution.

---

> ### Author Response · Authors · 2025-11-21
> **Response to the reproducibility**
>
> > Reproducibility
>
> We will upload the HuggingFace code soon, and we summarize the pseudo-code as below:
> - Here we illustrate the code of FuseNorm here:
>
>     ```
>     # post_attn_ln, post_ffn_ln are normalization layers
>     # attn is the attention module
>     # ffn is the feedforward or MoE module
>
>     def forward(x):
>         res = x
>
>         # Attention block
>         attn_out = attn(x)
>         y = post_attn_ln(attn_out + res)     # Post-Norm: Add then Norm
>
>         # FFN block
>         # Note: The input to FFN is y, but the residual added is still 'res'
>         ffn_out = ffn(y)
>         x = post_ffn_ln(ffn_out + res)  # Add original residual then Norm
>
>         return x
>
>     ```
>
> - PreNorm:
>     ```
>     # pre_attn_ln, pre_ffn_ln are normalization layers
>     # attn is the attention module
>     # ffn is the feedforward or MoE module
>
>     def forward(x):
>
>         # Attention block
>         res = x
>         norm_x = pre_attn_ln(x)
>         attn_out = attn(norm_x)
>         y = attn_out + res
>
>         # FFN block
>         res = y
>         norm_y = pre_ffn_ln(y)
>         ffn_out = ffn(norm_y)
>         x = ffn_out + res
>
>         return x
>
>     ```
>
> - LayerNorm Scale (built on PreNorm):
>     ```
>     # pre_attn_ln, pre_ffn_ln are normalization layers
>     # attn is the attention module
>     # ffn is the feedforward or MoE module
>
>     def forward(x):
>
>         # Attention block
>         res = x
>         norm_x = pre_attn_ln(x)
>         scale_factor = 1 / math.sqrt(cur_layer_index + 1) # here cur_layer_index begins from 0.
>         norm_x = norm_x * scale_factor
>         attn_out = attn(norm_x)
>         y = attn_out + res
>
>         # FFN block
>         res = y
>         norm_y = pre_ffn_ln(y)
>         norm_y = norm_y * scale_factor
>         ffn_out = ffn(norm_y)
>         x = ffn_out + res
>
>         return x
>
>     ```
>
> - Peri-LN (Gemma2-style):
>     ```
>     # pre_attn_ln, pre_ffn_ln, post_attn_ln, post_ffn_ln are normalization layers
>     # attn is the attention module
>     # ffn is the feedforward or MoE module
>
>     def forward(x):
>
>         # Attention block
>         res = x
>         norm_x = pre_attn_ln(x)
>         attn_out = attn(norm_x)
>         norm_attn_out = post_attn_ln(attn_out)
>         y = norm_attn_out + res
>
>         # FFN block
>         res = y
>         norm_y = pre_ffn_ln(y)
>         ffn_out = ffn(norm_y)
>         norm_ffn_out = post_ffn_ln(ffn_out)
>         x = norm_ffn_out + res
>
>         return x
>
>     ```
> - OLMO2 (reorder norm with QK-Norm):
>     ```
>     # QK_ln, post_attn_ln, post_ffn_ln are normalization layers
>     # attn_with_QK_Norm is the attention module which normalize the query and key at the head hidden dimension.
>     # ffn is the feedforward or MoE module
>     # merge_proj and output_proj are linear transformations.
>
>     def forward(x):
>
>         # Attention block
>         res = x
>         attn_out = attn_with_QK_Norm(x)
>         norm_attn_out = post_attn_ln(attn_out)
>         y = norm_attn_out + res
>
>         # FFN block
>         res = y
>         ffn_out = ffn(y)
>         norm_ffn_out = post_ffn_ln(ffn_out)
>         x = norm_ffn_out + res
>
>         return x
>
>     def attn_with_QK_Norm(x):
>
>         query, key, value = merge_proj(x) # separate the hidden size to head dimensions
>         norm_qurey = q_ln(query)
>         norm_key = k_ln(key)
>
>         attn_map = matmul(norm_query, Transform(norm_key)) / sqrt(head_dim)
>
>         # same with standard attn in the following computations.
>         attn_map = attn_map + attn_mask
>
>         attn_out = matmul(attn_map, value)
>
>         attn_out = output_proj(attn_out)
>
>         return attn_out
>     ```

---

> ### Author Response · Authors · 2025-11-27
> **Looking forward to further discussions!**
>
> Dear Reviewer Deba
>
> We sincerely appreciate the time and effort you've dedicated to reviewing our paper. We understand that you have a busy schedule, and we are truly grateful for your valuable feedback. As the Author-Reviewer discussion phase approaches its end, we are eager to know whether our response has addressed your concerns and if there are any additional questions or points you'd like to discuss. We would greatly appreciate the opportunity to engage in further discussion if needed. Thank you once again for your thoughtful consideration.
>
> Best regards,
>
> All Authors

---

### Official Review · Reviewer_hEpt · 2025-10-30

**Soundness:** 1
**Presentation:** 2
**Contribution:** 1
**Rating:** 2
**Confidence:** 5

**Summary:**

This paper proposes to achieve a best of both worlds between PreNorm and PostNorm in the Transformer architecture, by modifying the skip-connection/LayerNorm placements. Specifically, in a PostNorm architecture, the authors remove the skip connection from the FFN block, instead replacing it with a longer skip connection across both Attention and FFN blocks. The authors show improvements compared to PreNorm on multiple model sizes, and compared to some prior baselines.

**Strengths:**

1. The paper shows improvements over PreNorm architecture in pre-training across multiple model sizes, and over some prior works.
1. The proposed method achieves improved representation diversity across layers

**Weaknesses:**

1. A lot of the derivations are extremely handwavy, with numerous approximations and assumptions throughout in the derivations. ( E.g. $\approx$ occurs 15 times in the manuscript.).
1. Section 4 of the paper, covering depth and width scaling, are not proposing anything new, and are extremely redundant. For example, the  same/similar scaling has often been proposed before (e.g., see section K.3 of https://arxiv.org/pdf/2403.09635), and the "width scaling" is the authors simply verifying that prior works for LR transfer work with their architecture as well.
1. In section 5, the authors explicitly admit to an incomplete baseline study. A claim to "consistently outperform...advanced variants" is premature unless all baselines are working correctly.
1. In section 6, the authors point to their architecture being trainable with higher LR than PostNorm as proof of "mitigated gradient decay", but this conclusion does not necessarily hold. For example, a model with reduced sensitivity (eg. a linear layer with very large param values, followed by a layernorm) will be trainable with a much larger LR, but this does not say anything about inherent stability or instability of that model.
1. Even assuming all the derivations of the authors are correct, their proposed method only allows training 2x deeper post-norm models (equation 11). Prior works such as DeepNorm instead extend it to 100s of layers.

**Questions:**

1. Was the learning rate and initialization hyper-parameters set from known-good values from prior works, or perhaps hyper-parameter searched for all the baselines in Table 21? Inefficient setting of these can significantly affect performance.
1. (minor, no author rebuttal needed) The pdf seems to be somewhat bugged - searching and highlighting text is broken in several pages across multiple pdf readers. I have not observed the same in other papers.

---

> ### Author Response · Authors · 2025-11-21
> **Response to W1 and W2**
>
> > W1: A lot of the derivations are extremely handwavy, with numerous approximations and assumptions throughout in the derivations. ( E.g. occurs 15 times in the manuscript.).
>
> Thank you for your valuable suggestion on this issue. While this issue is mainly caused by the approximation in Equation 9, and we have already tried to explain this in footnote 1. But we are working on this issue and hope to provide a much stronger version soon.
>
> > W2: Section 4 of the paper, covering depth and width scaling, are not proposing anything new, and are extremely redundant. For example, the same/similar scaling has often been proposed before (e.g., see section K.3 of https://arxiv.org/pdf/2403.09635), and the "width scaling" is the authors simply verifying that prior works for LR transfer work with their architecture as well.
>
> Thank you for pointing this out. Our original intent was to provide the necessary background and demonstrate that FuseNorm requires specific initialization constraints (aligned with standard Megatron initialization) to avoid training instability. We view this as a necessary condition for valid large-scale training, rather than a novel contribution in itself. We can successfully train a good FuseNorm counterpart without dealing with any gradient vanishing or exploding problems. Such 'once scaling' is quite necessary for industrial scaling, since it is also difficult for companies to ablation all hyper-parameters on a large-scale model.
>
> We are glad to see that other reviewers (pvm3) found the theoretical guidelines in this section practical for stable training. However, we agree that the presentation can be improved. We will significantly condense the background information in Section 4 and explicitly clarify that we are adopting existing scaling laws to ensure FuseNorm's robust optimization.

---

> ### Author Response · Authors · 2025-11-21
> **Response to W3**
>
> > In section 5, the authors explicitly admit to an incomplete baseline study. A claim to "consistently outperform...advanced variants" is premature unless all baselines are working correctly.
>
> We have already added the missing baselines that all reviewers have raised. We hope the comparison now and the newly added challenging setting could address your concerns! The results are summarized in the following table:
>
> | Model                      | Wiki.(ppl) | LMB.(ppl) | LMB.     | PIQA     | Hella.   | SCIQ     | ARC-c    | Winogrande| Avg.     |
> | -------------------------- | ---------- | --------- | -------- | -------- | -------- | -------- | -------- | --------  | -------- |
> | PreNorm (lr=2e-4)          | 12.5       | 6.9       | 61.4     | 73.7     | 64.4     | 90.7     | 34.4     | 60.1      | 64.1     |
> | FuseNorm (old)             | 11.7       | 6.4       | 63.9     | 75.8     | 67.2     | 91.6     | 35.2     | 61.5      | 65.9     |
> | FuseNorm (lr=2e-4)         | 11.8       | 5.8       | 64.2     | 75.7     | 66.9     | 92.0     | 36.3     | 64.2      | **66.6** |
> | Mix-LN (lr=2e-4)           | 12.3       | 6.9       | 61.3     | 75.6     | 65.1     | 90.1     | 34.1     | 61.4      | 64.6     |
> | HybridNorm (lr=1e-4)       | 12.1       | 6.5       | 62.7     | 75.1     | 66.2     | 91.4     | 35.4     | 61.3      | 65.4     |
> | LayerNorm Scale (lr=2e-4)  | 13.6       | 14.9      | 51.0     | 73.1     | 60.7     | 86.1     | 32.2     | 58.3      | 60.2     |
> | Peri-LN (lr=2e-4)          | 12.9       | 7.9       | 58.2     | 73.8     | 62.7     | 90.8     | 32.1     | 60.3      | 63.0     |
> | Peri-LN (lr=1e-4)          | 12.3       | 7.2       | 59.5     | 75.7     | 65.1     | 90.3     | 35.2     | 59.9      | 64.3     |
> | OLMO2 (lr=2e-4)            | 14.0       | 9.9       | 53.7     | 73.1     | 59.2     | 87.4     | 30.6     | 57.2      | 60.2     |
>
> For a fairer comparison, we follow the suggestion of Reviewer pvm3, using a learning rate of 2e-4 to rerun our FuseNorm. Here, we removed the width scaling strategy, which means all initializations are totally the same! Besides, we add the results of Peri-LN (which is Gemma2-style architectures, namely sandwich-like norm), OLMO2, and LayerNorm Scale to show a comprehensive comparison. Through the updated results, we summarized that:
>
> - Superior Performance of FuseNorm: Under these strictly controlled settings, FuseNorm (lr=2e-4) achieves the best overall performance. It reaches an average accuracy of 66.6, surpassing our originally reported results by 0.7 (stronger results on Lambada, SCIQ, ARC-C, and Winogrande). Notably, it significantly outperforms the PreNorm baseline by 2.5 average score, which is indeed significant.
> - Optimized Baselines for Fair Comparison: Our goal is to compare FuseNorm against the best possible version of each baseline. Therefore, we tuned the hyperparameters for the competing methods:
>     - HybridNorm: We observed gradient vanishing with learning rates of 2e-4 and 1.5e-4; thus, we report the optimal stable result at 1e-4.
>     - Mix-LN: We performed a trade-off analysis, finding that a configuration of 16 PostNorm layers and 48 PreNorm layers yielded the lowest training loss.
>     - Peri-LN: We found that a 1e-4 learning rate yielded significantly better convergence and performance than 2e-4 for the Gemma-2 style architecture.
> - Investigation of OLMO2 and LayerNorm Scale: We are currently conducting further hyperparameter tuning for OLMO2 and LayerNorm Scale. However, preliminary analysis suggests reasons for their current lower performance compared to PreNorm:
>     - LayerNorm Scale: As the model depth increases (64 layers), the learned scaling factors may become extremely small, potentially hindering signal propagation. The original paper did not validate this method at this depth, suggesting it may require specific tuning for deep networks.
>     - OLMO2: While we utilized the OLMO2 architecture (including QK-Norm and reorder-norm), we maintained the standard Megatron initialization for fairness, rather than the fixed standard deviation (0.02) used in their technical report. We are currently validating the impact of this initialization difference.
>
> - Overall: These updated experiments demonstrate that FuseNorm is robust and achieves the best performance when compared fairly against a wide range of optimized baselines.

---

> ### Author Response · Authors · 2025-11-21
> **Response to W4, W5 and Q1:**
>
> > W3: In section 6, the authors point to their architecture being trainable with higher LR than PostNorm as proof of "mitigated gradient decay", but this conclusion does not necessarily hold. For example, a model with reduced sensitivity (eg. a linear layer with very large param values, followed by a layernorm) will be trainable with a much larger LR, but this does not say anything about inherent stability or instability of that model.
>
> Our intuition is not to highlight that our FuseNorm can benefit from a larger learning rate. While we would like to show that with the standard but a proper megatron initialization, FuseNorm can be expanded with no other burden, even in the industrial environment. In this way, we can omit the unnecessary hyper-parameter adjustments and can successfully train a good model once. While, we also add the new results of our FuseNorm, which was trained without width scaling (using learning rate as 2e-4, the same with PreNorm baselines). We see FuseNorm behaves much better than the previous results (avg score 66.6 vs 65.9).
>
>
> > W5: Even assuming all the derivations of the authors are correct, their proposed method only allows training 2x deeper post-norm models (equation 11). Prior works such as DeepNorm instead extend it to 100s of layers
>
>
> We clarify a misunderstanding regarding Equation 11 and provide empirical evidence demonstrating that FuseNorm scales successfully to **512 layers**, matching or exceeding the capabilities of benchmarks like DeepNorm.
>
> 1. Clarification on Equation 11 vs. Our Full Method:
>
>     - Eq. 11 ($G_{Post} \approx G_{Fuse}(2L)$) implies FuseNorm has a naturally slower gradient decay than PostNorm. However, this equation compares the architectures under standard initialization to highlight FuseNorm's inherent structural advantage. It is not the theoretical limit of our method.
>
>     - As detailed in Theorem 1 (Section 4.1), our full method incorporates a principled Scale Init strategy (scaling weights by $\mathcal{O}(1/\sqrt{L})$). This ensures that the signal variance scales as $\mathcal{O}(1/L)$, theoretically guaranteeing stable gradients for arbitrary depths at initialization, similar to the scaling principles used in DeepNorm.
>
> 2. Empirical Proof: Scaling to 512 Layers (New Stress-Test)
>
>     To experimentally refute the "2x limit," we conducted a stress test by training FuseNorm models with exponentially increasing depths from 32 to 512 layers (see Figure 14 in the Appendix).
>     - Setups: We used the same hidden size (384) and also adopted the Scale Init strategy.  Crucially, we maintained a constant learning rate (2 \times 10^{-4}) across all depths (from 32 to 512).
>     - "Deeper is Better": FuseNorm trains stably even at 512 layers with a strict monotonic decrease in training loss. This confirms that FuseNorm avoids depth degradation and scales effectively to hundreds of layers.
>
>
> > Q1: Was the learning rate and initialization hyper-parameters set from known-good values from prior works, or perhaps hyper-parameter searched for all the baselines in Table 21? Inefficient setting of these can significantly affect performance.
>
> In the updated version, all models share the same initialization strategy. For the learning rate, for a fair comparison, we used $2e^{-4}$ as the default setting. We also carefully test the known-good values from previous studies and report the results within different learning rates. As discussed in the response to W3, we tuned other baselines at several learning rates to ensure a strong enough conclusion.

---

> > ### Comment · Reviewer_hEpt · 2025-11-24
> > **Rebuttal Acknowledgement**
> >
> > Thank you for the rebuttal response. I will maintain my score for now.

---

> > > ### Author Response · Authors · 2025-11-27
> > > **Thanks for the response**
> > >
> > > Dear Reviewer hEpt
> > >
> > > We sincerely appreciate the time and effort you've dedicated to reviewing our paper.
> > >
> > > We have conducted extensive experiments and comprehensive analysis to strengthen our conclusions and improve the manuscript. If our response does not adequately address your concerns, we welcome further questions or clarifications on any unresolved issues. We view this as an opportunity to refine our work. Thank you for your thoughtful review, and we look forward to your feedback.
> > >
> > > Best regards,
> > >
> > > All Authors

---

### Official Review · Reviewer_icoP · 2025-10-31

**Soundness:** 3
**Presentation:** 3
**Contribution:** 2
**Rating:** 6
**Confidence:** 4

**Summary:**

The paper introduces FuseNorm, a normalization design that keeps a direct input–output skip connection through each Transformer block while applying normalization after both the attention and MLP sublayers. This layout aims to combine the gradient stability of PreNorm and the representational strength of PostNorm. The authors analyze its gradient and variance behavior, propose a depth-dependent initialization scheme, and validate the approach on dense and MoE models up to 16B parameters. FuseNorm yields small but consistent gains over PreNorm and maintains stable training where PostNorm becomes unstable.

**Strengths:**

* The method is simple, clearly defined, and easy to integrate into existing architectures. While training stability for large-scale Transformers has been extensively studied in recent months through various architectural innovations, the proposed method still contributes a refreshing perspective to this ongoing line of work.
 * Theoretical and empirical analyses are coherent: the scaling rule derived from variance control matches practical stability trends.
 * Results are consistent across model sizes and architectures, with diagnostic evidence (gradient norms, inter-layer similarity) that supports the claims.

**Weaknesses:**

* The paper omits comparisons to recent normalization schemes such as Peri-LN (arXiv:2502.02732), which has been adopted by several major LLMs and addresses the same issues. The paper’s motivation and positioning substantially overlap with prior works such as MixLN and HybridLN, which limits its perceived novelty.
 * The reported improvements are modest relative to the added complexity.

**Questions:**

- How does FuseNorm compare with recently proposed peri- or sandwich-style normalization schemes such as Peri-LN under identical hyperparameters and compute budgets?

---

> ### Author Response · Authors · 2025-11-21
> **Response to W1 and Q1 -- PART 1**
>
> Thank you for raising this point, which is also a common concern! Our comparisons with previous work were conducted on our largest configuration, and due to computational constraints, we were unable to complete all experiments before the submission deadline. Regarding the baselines, we did re-implement several, including OLMO2 and LayerNorm-Scale. Initially, our reproduced results were lower than those reported in their original papers (while the configuration here is slightly different). We omitted them to avoid confusion, but subsequent verification confirmed our implementation was correct. We have since re-run the OLMO2 experiment with a 2e-4 learning rate and added an embedding norm for more robust training, and we have included these updated results.
>
> | Model                      | Wiki.(ppl) | LMB.(ppl) | LMB.     | PIQA     | Hella.   | SCIQ     | ARC-c    | Winogrande| Avg.     |
> | -------------------------- | ---------- | --------- | -------- | -------- | -------- | -------- | -------- | --------  | -------- |
> | PreNorm (lr=2e-4)          | 12.5       | 6.9       | 61.4     | 73.7     | 64.4     | 90.7     | 34.4     | 60.1      | 64.1     |
> | FuseNorm (old)             | 11.7       | 6.4       | 63.9     | 75.8     | 67.2     | 91.6     | 35.2     | 61.5      | 65.9     |
> | FuseNorm (lr=2e-4)         | 11.8       | 5.8       | 64.2     | 75.7     | 66.9     | 92.0     | 36.3     | 64.2      | **66.6** |
> | Mix-LN (lr=2e-4)           | 12.3       | 6.9       | 61.3     | 75.6     | 65.1     | 90.1     | 34.1     | 61.4      | 64.6     |
> | HybridNorm (lr=1e-4)       | 12.1       | 6.5       | 62.7     | 75.1     | 66.2     | 91.4     | 35.4     | 61.3      | 65.4     |
> | LayerNorm Scale (lr=2e-4)  | 13.6       | 14.9      | 51.0     | 73.1     | 60.7     | 86.1     | 32.2     | 58.3      | 60.2     |
> | Peri-LN (lr=2e-4)          | 12.9       | 7.9       | 58.2     | 73.8     | 62.7     | 90.8     | 32.1     | 60.3      | 63.0     |
> | Peri-LN (lr=1e-4)          | 12.3       | 7.2       | 59.5     | 75.7     | 65.1     | 90.3     | 35.2     | 59.9      | 64.3     |
> | OLMO2 (lr=2e-4)            | 14.0       | 9.9       | 53.7     | 73.1     | 59.2     | 87.4     | 30.6     | 57.2      | 60.2     |
>
> For a fairer comparison, we follow the suggestion of Reviewer pvm3, using a learning rate of 2e-4 to rerun our FuseNorm. Here, we removed the width scaling strategy, which means all initializations are totally the same! Besides, we add the results of Peri-LN (which is Gemma2-style architectures, namely sandwich-like norm), OLMO2, and LayerNorm Scale to show a comprehensive comparison. Through the updated results, we summarized that:
>
> - Superior Performance of FuseNorm: Under these strictly controlled settings, FuseNorm (lr=2e-4) achieves the best overall performance. It reaches an average accuracy of 66.6, surpassing our originally reported results by 0.7 (stronger results on Lambada, SCIQ, ARC-C, and Winogrande). Notably, it significantly outperforms the PreNorm baseline by 2.5 average score, which is indeed significant.
> - Optimized Baselines for Fair Comparison: Our goal is to compare FuseNorm against the best possible version of each baseline. Therefore, we tuned the hyperparameters for the competing methods:
>     - HybridNorm: We observed gradient vanishing with learning rates of 2e-4 and 1.5e-4; thus, we report the optimal stable result at 1e-4.
>     - Mix-LN: We performed a trade-off analysis, finding that a configuration of 16 PostNorm layers and 48 PreNorm layers yielded the lowest training loss.
>     - Peri-LN: We found that a 1e-4 learning rate yielded significantly better convergence and performance than 2e-4 for the Gemma-2 style architecture.
> - Investigation of OLMO2 and LayerNorm Scale: We are currently conducting further hyperparameter tuning for OLMO2 and LayerNorm Scale. However, preliminary analysis suggests reasons for their current lower performance compared to PreNorm:
>     - LayerNorm Scale: As the model depth increases (64 layers), the learned scaling factors may become extremely small, potentially hindering signal propagation. The original paper did not validate this method at this depth, suggesting it may require specific tuning for deep networks.
>     - OLMO2: While we utilized the OLMO2 architecture (including QK-Norm and reorder-norm), we maintained the standard Megatron initialization for fairness, rather than the fixed standard deviation (0.02) used in their technical report. We are currently validating the impact of this initialization difference.
>
> - Overall: These updated experiments demonstrate that FuseNorm is robust and achieves the best performance when compared fairly against a wide range of optimized baselines.

---

> ### Author Response · Authors · 2025-11-21
> **Response to W1 and Q1 -- PART 2**
>
> > More experiments on a 128-layer setting
>
> For more convincing results to show the potential of FuseNorm, we also conducted experiments on a configuration: {hidden size: 2048, layers: 128, intermediate size 5632}. Note that, to keep a totally fair comparison, we search for the best learning rate on PreNorm models, and set 1.2e-4 as the default setting for all models. We trained these models up to 400B tokens for robust conclusions. The results are summarized as below:
>
> | Model                      | Wiki.(ppl) | LMB.(ppl) | LMB.     | PIQA     | Hella.   | SCIQ     | ARC-c    | Winograde| Avg.     |
> | -------------------------- | ---------- | --------- | -------- | -------- | -------- | -------- | -------- | -------- | -------- |
> | PreNorm (lr=1.2e-4)        | 11.9       | 6.6       | 62.6     | 75.3     | 65.2     | 90.5     | 32.6     | 60.1     | 64.4     |
> | FuseNorm (lr=1.2e-4)       | 10.4       | 5.5       | 65.4     | 77.5     | 70.6     | 92.6     | 37.7     | 66.6     | 68.4     |
> | Mix-LN  (lr=1.2e-4)        | 11.4       | 6.2       | 64.0     | 76.1     | 67.5     | 90.4     | 35.7     | 62.5     | 66.0     |
> | HybridNorm (lr=1.2e-4)     | failed     | failed    | failed   | failed   | failed   | failed   | failed   | failed   | failed   |
>
> - We find that a 6.5B 128-layer PreNorm suffers from 'depth degradation' issue, it just shows similar performance with its 5B counterpart trained by 200B tokens.
> - FuseNorm significantly outperforms PreNorm (up to 4 average points) and other baselines, including Mix-LN and HybridNorm. Under this setting, Mix-LN is still stronger than the PreNorm, but HybridNorm often faces with gradient vanishing problem, even when reducing the learning rate to 6e-5.
>
> > About the novelty
>
> For the novelty issue, we kindly refer you to our response to the Reviewer iuA5. We think our work is a good complement to the current investigation in the literature.

---

> ### Author Response · Authors · 2025-11-21
> **Response to W2: about the complexity**
>
> > Reported improvements are modest relative to the added complexity.
>
> Our computation complexity is totally the same as PreNorm, where we employ an additional embedding norm instead of the final norm in PreNorm. There are two normalization functions at each layer, which are the same as PreNorm. We also plot the model architecture in Figure 1

---

### Official Review · Reviewer_pvm3 · 2025-11-08

**Soundness:** 2
**Presentation:** 2
**Contribution:** 2
**Rating:** 4
**Confidence:** 4

**Summary:**

This paper introduces FuseNorm, a normalization strategy that unifies the training stability of Pre-LayerNorm and the performance advantages of Post-LayerNorm in Transformer architectures. The key idea is to retain a residual path while applying a Post-LN-style normalization at the block output, thereby resetting variance and maintaining balanced gradients across depth. Theoretical analysis shows that FuseNorm preserves stable Jacobian spectra, and empirical studies confirm consistent stability and performance improvements across both dense and Mixture-of-Experts models.

**Strengths:**

- The architectural modification is simple and intuitive, replacing the FFN residual with the original block input and normalizing the final output.
- The theoretical condition for depth scaling provides a practical guideline for stable training.
- Empirical results show consistent improvement trends across dense and MoE settings, suggesting potential practical value.

**Weaknesses:**

1. **Figures and visual evidence are unconvincing and inconsistently presented.** Many of the figures (Figs. 2–7) lack consistent axis ranges, normalization, and clarity in what they aim to demonstrate, making cross-figure interpretation difficult. For example, Figure 3 is intended to illustrate training collapse without scaling, yet it only shows three layer traces (layers 1, 12, 24) from a single run, without comparison to Pre-, Post-, or scaled FuseNorm variants. This makes it impossible to assess generality, effect size, or reproducibility. A unified plot contrasting “no-scale vs. scale” across normalization types with mean ± std over multiple seeds is needed. Moreover, Figure 6—claimed to demonstrate severe representation collapse in Pre-LN and its prevention by FuseNorm—does not visually support such a conclusion. When inspecting the heatmaps, the difference between Pre-LN and FuseNorm appears marginal, with large red regions (high similarity) persisting in both cases. The improvement is not visually evident enough to justify the strong qualitative statement in the text.
Each figure and caption should clearly specify the experimental setup, number of runs, normalization scheme, and intended takeaway, ensuring visual evidence aligns with the claimed phenomena.

2. **Missing strong baselines, fairness issues, and lack of comparison to Gemma-style architectures.** The paper lacks head-to-head results with strong baselines such as DeepNorm and NormFormer, both of which explicitly target the same stability–performance trade-off. In addition, the OLMO2 and LayerNorm-Scaling results are reported as “being re-run,” raising questions about hyperparameter fairness. A rigorous comparison should fix dataset, token count, learning-rate schedule, initialization, and random seeds (≥ 3), with clearly defined criteria for “training failure.” Furthermore, recent open-source models such as Gemma 2/3 employ a Peri-LN-style normalization [1], which shares similar motivations of balancing gradient flow and variance growth. A theoretical and empirical comparison between FuseNorm and such Peri-LN architectures would substantially strengthen the paper’s positioning and clarify whether the proposed method provides distinct or complementary benefits. Adding Peri-LN to Table 2, along with gradient-norm profiles or stability-transfer plots (Fig. 4), would make the evaluation more comprehensive and fair.

3. **Ablation insufficiency – unclear contribution of structure vs. initialization.**  The current results do not isolate whether the observed improvements stem from the new FuseNorm block design itself or from the “Scale-Init” strategy introduced for depth scaling.

4. **Missing quantitative validation of theoretical claims.** The theoretical analysis predicts that the variance of each residual branch should scale and that FuseNorm reduces gradient-decay rates relative to Post-LN. However, the paper provides no empirical evidence verifying these relationships.


[1] Kim et al. "Peri-ln: Revisiting normalization layer in the transformer architecture." ICML2025.

**Questions:**

All major questions and requests for clarification are already integrated into the Weaknesses section for clarity and conciseness.

---

> ### Author Response · Authors · 2025-11-21
> **Response to W1**
>
> > Figures and visual evidence are unconvincing and inconsistently presented.
>
> We sincerely thank you for the rigorous feedback. We agree that the original visual presentation lacked sufficient consistency and comparative depth. To address this, we have thoroughly revised all figures (Figs. 2–7) and added substantial new experimental evidence to ensure rigor, clarity, and reproducibility.
>
> 1. Addressing Inconsistency and Clarity (Figs. 2–7):
>
>     - Unified Visual Standards: We have standardized axis ranges across related experiments to facilitate cross-figure interpretation.
>
>     - Early Dynamics (Figs. 2, 3, 4): We unified the X-axis to the first 5,000 training steps for these figures. This consistent window allows for a direct, cross-figure assessment of initialization impacts and early-stage stability (e.g., linking the divergence in Fig. 2 directly to the gradient collapse in Fig. 3 within the exact same timeframe).
>     - Full Training (Fig. 5 & New Fig. 13): For performance comparisons, we standardized the X-axis to "Training Tokens" and used comparable Loss Y-axis ranges to demonstrate long-term convergence behavior.
>
>     - Detailed Captions: We have rewritten all captions to strictly follow the structure: (1) Experimental Setup (specifying model width/depth, steps, seeds), (2) Normalization Scheme, and (3) Key Takeaway. For example, in Figure 4, we explicitly state the fixed depth ($L=64$) and specific width scaling ($d=640 \to 2560$) to clarify the experimental design.
>
>
>
> 2. Major Overhaul of Figure 3 (Gradient Dynamics):
>
>     We acknowledge the reviewer's valid point that the original Figure 3 was insufficient. We have replaced it with a comprehensive analysis (see Figure 13 in Appendix).
>
>     - Full Visibility: Instead of 3 layers, we now visualize gradient norms for all 24 layers (L0–L23).
>     - Rigorous Comparison: We explicitly compare PreNorm (top) vs. FuseNorm (bottom) and Global Init (Red) vs. Scale Init (Blue).
>     - Reproducibility: We plot the gradient norm trajectories from 3 independent random seeds. The overlapping traces demonstrate highly consistent behavior, cconfirming that the behavior is consistent across seeds and not an artifact of a single run.
>     - Visual Proof: The new plot provides undeniable evidence that FuseNorm with Global Init suffers from catastrophic collapse in early layers (norms $< 10^{-12}$), while Scale Init fully stabilizes the gradients, matching the robust profile of PreNorm.
>
>
>
> 3. Rectifying Figure 6 (Visual & Quantitative Evidence of Collapse):
>
>     We appreciate this observation. We clarify that the persistent red regions along the diagonal are expected, reflecting the necessary local continuity in residual networks. The critical indicator of collapse is not local similarity, but the lack of long-range feature evolution.To demonstrate this effectively, we provided below evidence:
>
>     - Quantitative Proof (New Figure 12): We plotted the *Average Similarity vs. Layer Distance* ($|L_i - L_j|$). This plot clearly disentangles the "local" from the "long-range":
>         - PreNorm: The curve flattens out, maintaining high similarity ($>0.5$) even at maximum layer distance. This quantitatively confirms representation collapse—the network fails to evolve features significantly from input to output.
>         - FuseNorm: The curve shows a rapid, linear decay down to $\approx 0.25$. This wide gap proves that FuseNorm drives significant long-range feature evolution, effectively mitigating collapse.
>     - Enhanced Heatmap Visualization: To make this long-range decorrelation visually distinct from the "local red noise," we rescaled the heatmap colors to $[0.5, 1.0]$. The revised Figure 6 now clearly displays a deep-blue region in FuseNorm's off-diagonal, visually corroborating the quantitative decay shown in Figure 12.
>
>     - New Evidence of Generality and Scalability: To assess generality and effect size as requested, we added two "stress-test" analysis:
>
>         - 128-Layer/6.5B Comparison (Figure 9): We demonstrate that at extreme depths (128 layers), baselines like PostNorm and HybridNorm fail to converge, while **FuseNorm significantly outperforms PreNorm and Mix-LN over 400B tokens**.
>
>         - "Deeper is Better" (Figure 14): We visualize loss curves for models scaling from 32 to 512 layers using a fixed learning rate ($2\times10^{-4}$), showing monotonic performance improvement without instability.

---

> ### Author Response · Authors · 2025-11-21
> **Response to W2 -- PART 1**
>
> Thank you for raising this point. Our comparisons with previous work were conducted on our largest configuration, and due to computational constraints, we were unable to complete all experiments before the submission deadline. Regarding the baselines, we did re-implement several, including OLMO2 and LayerNorm-Scale. Initially, our reproduced results were lower than those reported in their original papers (while the configuration here is slightly different). We omitted them to avoid confusion, but subsequent verification confirmed our implementation was correct. We have since re-run the OLMO2 experiment with a 2e-4 learning rate and added an embedding norm for more robust training, and we have included these updated results.
>
> | Model                      | Wiki.(ppl) | LMB.(ppl) | LMB.     | PIQA     | Hella.   | SCIQ     | ARC-c    | Winogrande| Avg.     |
> | -------------------------- | ---------- | --------- | -------- | -------- | -------- | -------- | -------- | --------  | -------- |
> | PreNorm (lr=2e-4)          | 12.5       | 6.9       | 61.4     | 73.7     | 64.4     | 90.7     | 34.4     | 60.1      | 64.1     |
> | FuseNorm (old)             | 11.7       | 6.4       | 63.9     | 75.8     | 67.2     | 91.6     | 35.2     | 61.5      | 65.9     |
> | FuseNorm (lr=2e-4)         | 11.8       | 5.8       | 64.2     | 75.7     | 66.9     | 92.0     | 36.3     | 64.2      | **66.6** |
> | Mix-LN (lr=2e-4)           | 12.3       | 6.9       | 61.3     | 75.6     | 65.1     | 90.1     | 34.1     | 61.4      | 64.6     |
> | HybridNorm (lr=1e-4)       | 12.1       | 6.5       | 62.7     | 75.1     | 66.2     | 91.4     | 35.4     | 61.3      | 65.4     |
> | LayerNorm Scale (lr=2e-4)  | 13.6       | 14.9      | 51.0     | 73.1     | 60.7     | 86.1     | 32.2     | 58.3      | 60.2     |
> | Peri-LN (lr=2e-4)          | 12.9       | 7.9       | 58.2     | 73.8     | 62.7     | 90.8     | 32.1     | 60.3      | 63.0     |
> | Peri-LN (lr=1e-4)          | 12.3       | 7.2       | 59.5     | 75.7     | 65.1     | 90.3     | 35.2     | 59.9      | 64.3     |
> | OLMO2 (lr=2e-4)            | 14.0       | 9.9       | 53.7     | 73.1     | 59.2     | 87.4     | 30.6     | 57.2      | 60.2     |
>
> For a fairer comparison, we follow the suggestion of Reviewer pvm3, using a learning rate of 2e-4 to rerun our FuseNorm. Here, we removed the width scaling strategy, which means all initializations are totally the same! Besides, we add the results of Peri-LN (which is Gemma2-style architectures, namely sandwich-like norm), OLMO2, and LayerNorm Scale to show a comprehensive comparison. Through the updated results, we summarized that:
>
> - Superior Performance of FuseNorm: Under these strictly controlled settings, FuseNorm (lr=2e-4) achieves the best overall performance. It reaches an average accuracy of 66.6, surpassing our originally reported results by 0.7 (stronger results on Lambada, SCIQ, ARC-C, and Winogrande). Notably, it significantly outperforms the PreNorm baseline by 2.5 average score, which is indeed significant.
> - Optimized Baselines for Fair Comparison: Our goal is to compare FuseNorm against the best possible version of each baseline. Therefore, we tuned the hyperparameters for the competing methods:
>     - HybridNorm: We observed gradient vanishing with learning rates of 2e-4 and 1.5e-4; thus, we report the optimal stable result at 1e-4.
>     - Mix-LN: We performed a trade-off analysis, finding that a configuration of 16 PostNorm layers and 48 PreNorm layers yielded the lowest training loss.
>     - Peri-LN: We found that a 1e-4 learning rate yielded significantly better convergence and performance than 2e-4 for the Gemma-2 style architecture.
> - Investigation of OLMO2 and LayerNorm Scale: We are currently conducting further hyperparameter tuning for OLMO2 and LayerNorm Scale. However, preliminary analysis suggests reasons for their current lower performance compared to PreNorm:
>     - LayerNorm Scale: As the model depth increases (64 layers), the learned scaling factors may become extremely small, potentially hindering signal propagation. The original paper did not validate this method at this depth, suggesting it may require specific tuning for deep networks.
>     - OLMO2: While we utilized the OLMO2 architecture (including QK-Norm and reorder-norm), we maintained the standard Megatron initialization for fairness, rather than the fixed standard deviation (0.02) used in their technical report. We are currently validating the impact of this initialization difference.
>
> - Overall: These updated experiments demonstrate that FuseNorm is robust and achieves the best performance when compared fairly against a wide range of optimized baselines.

---

> ### Author Response · Authors · 2025-11-21
> **Resonpse to W2 -- PART 2**
>
> For more convincing results to show the potential of FuseNorm, we also conducted experiments on a configuration: {hidden size: 2048, layers: 128, intermediate size 5632}. Note that, to keep a totally fair comparison, we search for the best learning rate on PreNorm models, and set 1.2e-4 as the default setting for all models. We trained these models up to 400B tokens for robust conclusions. The results are summarized as below:
>
> | Model                      | Wiki.(ppl) | LMB.(ppl) | LMB.     | PIQA     | Hella.   | SCIQ     | ARC-c    | Winograde| Avg.     |
> | -------------------------- | ---------- | --------- | -------- | -------- | -------- | -------- | -------- | -------- | -------- |
> | PreNorm (lr=1.2e-4)        | 11.9       | 6.6       | 62.6     | 75.3     | 65.2     | 90.5     | 32.6     | 60.1     | 64.4     |
> | FuseNorm (lr=1.2e-4)       | 10.4       | 5.5       | 65.4     | 77.5     | 70.6     | 92.6     | 37.7     | 66.6     | 68.4     |
> | Mix-LN  (lr=1.2e-4)        | 11.4       | 6.2       | 64.0     | 76.1     | 67.5     | 90.4     | 35.7     | 62.5     | 66.0     |
> | HybridNorm (lr=1.2e-4)     | failed     | failed    | failed   | failed   | failed   | failed   | failed   | failed   | failed   |
>
> - We find a 6.5B 128-layer PreNorm suffers from 'depth degradation' issue; it just shows similar performance with its 5B counterpart trained by 200B tokens.
> - FuseNorm significantly outperforms PreNorm and other baselines, including Mix-LN and HybridNorm. Under this setting, Mix-LN is still stronger than the PreNorm, but HybridNorm often faces with gradient vanishing problem, even when reducing the learning rate to 6e-5.

---

> ### Author Response · Authors · 2025-11-21
> **Response to W3 and W4**
>
> > W3: Ablation insufficiency
>
> As discussed in Section 4, our FuseNorm can face training spikes with global initialization. In our previous manuscript, we adopted depth scaling and width scaling to ensure robust performance. In the above experiments, we have shown that removing width scaling leads to no performance degradation when using the same initialization strategy with other baselines. On the other hand, the depth scaling is the default setting in vanilla Megatron initialization; thus, we used it for all baselines in our experiments.
>
> > W4: Missing quantitative validation of theoretical claims.
>
> We think this is also a similar concern to the above issue. We have already validated that without depth scaling, the gradient norm tends to vanish when the model goes deeper (See Figure 2 for details), thus we used the depth scaling strategy to address this issue.
>
> For the gradient norm decay problem, we kindly refer you to the results shown in Figure 7. We have already compared the gradient norm between our FuseNorm and PostNorm within different learning rate scenarios. We can clearly observe that our FuseNorm can show health gradient norms than PostNorm, especially at the bottom layers. We hope these results can address your concerns.

---

> ### Author Response · Authors · 2025-11-27
> **We are looking forward to further discussion!**
>
> Dear Reviewer pvm3
>
> We sincerely appreciate the time and effort you've dedicated to reviewing our paper. We understand that you have a busy schedule, and we are truly grateful for your valuable feedback. As the Author-Reviewer discussion phase approaches its end, we are eager to know whether our response has addressed your concerns and if there are any additional questions or points you'd like to discuss.
>
> In the current manuscript, we have clarified the captions you have suggested and updated the figures accordingly for better understanding. We have conducted additional experiments with/without width scaling and re-implemented other baselines carefully. We would greatly appreciate the opportunity to engage in further discussion if needed. Thank you once again for your thoughtful consideration.
>
> Best regards,
>
> All Authors

---

### Official Review · Reviewer_iuA5 · 2025-11-10

**Soundness:** 2
**Presentation:** 3
**Contribution:** 2
**Rating:** 4
**Confidence:** 4

**Summary:**

This paper introduces **FuseNorm**, a normalization strategy that integrates the advantages of both PreNorm and PostNorm while addressing their respective drawbacks. In particular,  FuseNorm leverages the training stability benefit of PreLN and combines it with the performance benefits of PostLN.

The authors provide theoretical justifications for how FuseNorm preventing representational collapse in deeper layers, while also improving gradient flow to counteract gradient decay during training.  Empirical results show consistent performance improvements over recent hybrid normalization schemes that attempt to mix or modify PreLN and PostLN strategies.

**Strengths:**

1. LayerNorm placement is still a very active research topic, with several recent works exploring position-specific variants like QK-, QKV-, and FFN-LayerNorm [1,2, 3], or hybrid approaches such as MixLN that aim to balance stability and performance in deeper models. This paper fits naturally into that discussion and and provide a unified and principled way to combine the strengths of existing normalization strategies


2. Authors have provided a clear mathematical analysis on how FuseNorm helps prevent representational collapse and mitigate gradient decay which offer an intuitive  understanding of normalization behavior in large and deep LLMs.

3. The experimental evaluation is thorough, comparing FuseNorm against recent hybrid normalization strategies across both dense and MoE-based FFN architectures, with model sizes ranging from 0.74B to 5B parameters. The performance gains in deeper LLMs is convincing for showing the utility of FuseNorm.


[1] Dehghani et al., Scaling vision transformers to 22 billion parameters, ICML 2023

[2] Zhuo et al., HybridNorm: Towards Stable and Efficient Transformer Training via Hybrid Normalization, 2025

[3]  Rybakov et al., Methods of improving llm training stability, 2024

**Weaknesses:**

1. The novelty of the work is quite limited. The idea of combining the benefits of PreLN and PostLN has already been well explored in prior works such as MixLN [1] and other hybrid normalization schemes. The proposed FuseNorm is an **incremental modification** rather than a fundamentally new concept, and the paper does not convincingly articulate what distinguishes it in terms of design principle or innovation.


2. The Authors  do not demonstrate how FuseNorm improves the **quality of internal representations**. Although it claims to prevent representational collapse, there’s no clear analysis---such as probing FFNs, or representation visualizations. Without showing how the internal representations, the paper’s claim about representational improvement is not convincing.  For example  eigen-value distribution or Rank-analysis shown in  [2,3].


[1] Li et al., Mix-LN: Unleashing the Power of Deeper Layers by Combining Pre-LN and Post-LN, ICLR 2025

[2] Loshchilov et al., nGPT: Normalized Transformer with Representation Learning on the Hypersphere, ICLR 2025

[3] Jha et al.,  Spectral Scaling Laws in Language Models: How Effectively Do Feed-Forward Networks Use Their Latent Space? EMNLP 2025

**Questions:**

Could the authors provide the eigenvalue distribution of FFN post-activations or weight matrices, or a layer-wise rank comparison between FuseNorm, PreLN, PostLN, and MixLN models? This would offer concrete evidence of how FuseNorm impacts internal representation quality and whether it genuinely prevents representational collapse in deeper layers.

---

> ### Author Response · Authors · 2025-11-21
> **Response to the novelty**
>
> We thank the reviewer for this important question, which was also raised by Reviewer icoP. We appreciate the opportunity to precisely differentiate our FuseNorm from prior art, including HybridNorm, Mix-LN, and Peri-LN. We apologize for the inadvertent omission of the Peri-LN reference; this was an oversight and will be corrected in the revised manuscript.
>
>
> The core motivation of FuseNorm lies from the foundational trade-off in Transformer normalization:
>
> - Pre-Norm: Offers stable training and a clear gradient path ("a clear path from bottom to the top"), which is highly effective for deep models. However, it can suffer from representation collapse as depth increases, where the residual branch dominates the update, weakening the model's expressive power.
>
> - Post-Norm: Generally produces stronger, more expressive representations. However, it is notoriously difficult to train due to unstable gradient flow, where bottom layers often face the gradient vanishing problem.
>
> The core novelty of FuseNorm is our attempt to unify these two advantages: We designed a method that maintains the stable training dynamics and gradient flow of Pre-Norm while simultaneously ensuring the strong representation fidelity of Post-Norm.
>
> Here, we would like to differentiate FuseNorm from prior work.
> While several methods have tried to solve this, their approaches are conceptually and mechanistically distinct from ours. We categorized these in our related work (Lines 103-107) and will expand this discussion:
>
> - vs. MixLN: This is an inter-layer (or block-level) approach. It structurally divides the network, using Post-Norm for bottom layers and Pre-Norm for top layers. In contrast, FuseNorm is an intra-layer method applied consistently within every block of the network.
>
> - vs. HybridNorm: This is the closest in intent (both are intra-layer), but our mechanism is different. HybridNorm applies a PreNorm-like normalization to the QKV projections and a separate Post-Norm to the FFN. We see this as applying two distinct normalization types. FuseNorm, however, is a single, unified mechanism designed to concurrently manage the information flow and the residual update.
>
> - vs. Peri-LN: This method uses a "sandwich" (or Gemma2-Norm) style, normalizing both the input and the output of a sub-layer. This is fundamentally a double-normalization strategy. FuseNorm is not a double-normalization method; it is a single, hybrid normalization step carefully designed to achieve the balance we described.
>
> In summary, while the goal of stable training and strong representation is shared, the high-level idea and technical execution of FuseNorm are novel. It is not an inter-layer schedule (like MixLN), nor a simple combination of existing norms (like HybridNorm), nor a double-normalization scheme (like Peri-LN). It is a distinct intra-layer design that fuses the benefits of Pre- and Post-Norm.

---

> ### Author Response · Authors · 2025-11-21
> **Response to W2: More analysis on internal representations and Question**
>
> We really thank you for your advice for further validating the quality of inner representations, and we conducted experiments using eigenvalue distribution and Rank-analysis, respectively. Corresponding results have been updated in the Appendix. Note that all experiments were done upon 5B dense models, covering PreNorm, FuseNorm, Mix-LN, and HybridNorm. Here, we would like to summarize several key findings:
>
> - Eigenvalue distribution: Following the setup in nGPT[1], we mainly plotted the sorted eigenvalues of the input embedding and the output embedding. As shown in **Figure 8**, we analyze the distribution of sorted eigenvalues normalized by their median value (log scale). While Mix-LN mitigates the extreme outliers seen in PreNorm, we observe that it suffers from a rapid decay in the tail spectrum, indicating potential dimensional collapse. In contrast, we demonstrate that FuseNorm maintains a significantly flatter and more uniform trajectory across the entire rank, ensuring a more isotropic representation space with higher effective capacity.
>
> - Condition number analysis: Moreover, we also plotted the condition number of MLP1 and MLP2 at each layer. As plotted in **Figure 10**, we can observe that FuseNorm demonstrates superior stability, **maintaining significantly lower condition numbers, particularly in the deeper layers**, which effectively prevents spectral degradation. In contrast, PreNorm exhibits elevated condition numbers in deeper blocks, reflecting the issue of deep layer degeneration where the representation capacity collapses. This comparison highlights FuseNorm's ability to preserve a well-conditioned and expressive latent space throughout the entire network depth.
>
>
> - Spectral utilization: Following the setup in [2], we evaluate the latent space efficiency using Hard Rank, Soft Rank, and Effective Dimension Ratio (EDR), as illustrated in **Figure 11**. We observe that standard PreNorm suffers from severe spectral collapse, averaging only 0.09 in Hard Rank. In contrast, FuseNorm consistently maintains superior expressivity. It achieves the highest Hard Rank (0.23) and Soft Rank (0.53), effectively balancing dominant and tail information. Notably, **FuseNorm dominates the composite EDR metric with a score of 30.66, which is more than doubling the capacity of PreNorm (12.99)**. This demonstrates that FuseNorm effectively mitigates spectral degeneration and preserves a rich representational space throughout the network depth.
>
> We have incorporated this discussion and analysis into our revised manuscript and cited the two suggested references. We hope these changes satisfactorily address your concerns, and we look forward to your further feedback
>
>
> [1] nGPT: Normalized Transformer with Representation Learning on the Hypersphere
> [2] Spectral scaling laws in language models: emphhow effectively do feed-forward networks use their latent space?

---

> ### Author Response · Authors · 2025-11-27
> **We are looking forward to further discussion!**
>
> Dear Reviewer iuA5
>
> We sincerely appreciate the time and effort you've dedicated to reviewing our paper. We understand that you have a busy schedule, and we are truly grateful for your valuable feedback. As the Author-Reviewer discussion phase approaches its end, we are eager to know whether our response has addressed your concerns and if there are any additional questions or points you'd like to discuss.
> In the current manuscript, we have added the analysis of internal representations and the corresponding reference. We would greatly appreciate the opportunity to engage in further discussion if needed. Thank you once again for your thoughtful consideration.
>
> Best regards,
>
> All Authors

---

### Author Response · Authors · 2025-11-25
**Global Response**

We are deeply grateful for the time and effort spent by all reviewers in evaluating our paper. Your feedbacks have guided us in executing a comprehensive revision of the manuscript, which includes the following significant improvements:

- Verified Robustness (See Table 1): We have rerun the FuseNorm experiments using a standard learning rate of $2e-4$, identical to the setting used for PreNorm and other baselines, and without relying on width scaling. The new results confirm that FuseNorm’s performance is robust and independent of specially tuned initialization methods.
- Expanded Comparative Analysis (See Table 2): We have incorporated additional, highly relevant baselines, including peri-LN, and explicitly integrated the results for previously explored methods such as OLMo2 and LayerNormScale, resulting in a more complete benchmark.
- New results of 128-layer setting (See Table 3): We conducted further, more rigorous experiments on a challenging 128-layer setting. This demonstrates the potential of FuseNorm to scale effectively and shows notable performance gains compared to PreNorm. The observation strongly confirms that FuseNorm successfully alleviates the representation collapse issue in deep networks.
- Enhanced Theoretical and Empirical Clarity: We have rigorously revised Section 3.3 to provide a more formal and precise theoretical analysis. The statements in Section 4 have also been revised for improved clarity and rigor.
- In-Depth Visual and Quantitative Analysis: We have updated Figure 5 and Figure 6, and added an extensive suite of new figures for deeper insights:
    - Figure 8: Analysis of internal representations (Eigenvalue of input embedding).
    - Figure 9: Training curves for the 128-layer setting.
    - Figure 10: Layer-wise condition number analysis.
    - Figure 11: Rank analysis.
    - Figure 12: Quantitative analysis specifically focused on representation collapse.


We are working on DeepNorm training and the Huggingface code, and we are looking forward to your discussions.

---

### Comment · Area_Chair_T8vG · 2025-11-28

Dear Reviewers,

Thank you for your time and efforts in serving as a reviewer. The authors have submitted their rebuttal, and this AC kindly asks you to review their response and assess whether your comments have been adequately addressed.

If you have not yet done so, please raise any remaining questions by adding comments and initiating discussion as needed for points that require further clarification.

ICLR encourages reviewers to actively engage in the discussion phase, so your prompt actions are especially valuable. Thank you very much for your continued efforts and valuable contributions.

Best regards,
Your AC

---

### Meta-Review · Area_Chair_fs4b · 2026-01-08

**Summary:**

The paper proposes "FuseNorm," a normalization strategy designed to balance the training stability of PreNorm with the performance advantages of PostNorm in Transformer architectures. Despite the authors' commendable efforts during the rebuttal phase to address missing baselines and clarify the presentation, the consensus leans towards rejection. The authors are encouraged to refine the theoretical grounding and perform a more rigorous ablation study isolating the architectural contributions for future submission.

**Reviewer Concerns:**

A primary concern raised by multiple reviewers  is that the proposed method appears to be an incremental modification within a crowded landscape of normalization variants. While the authors demonstrated that FuseNorm outperforms PreNorm in several settings, reviewers noted that the improvements are often modest. There is also a persistent concern regarding the isolation of benefits.

**Reviewer Scores:**

Some reviewers are positive while others tend to reject the paper. Reviewer hEpt did see the rebuttal and explicitly stated will maintain his score . Some of them fundamentally disagreed with the theoretical derivation  and the novelty. No amount of empirical data was likely to sway them from the opinion that the theoretical foundation was flawed.

---

### Decision · Program_Chairs · 2026-01-26

Reject